# Behavior Injection: Preparing Language Models for Reinforcement Learning

**Zhepeng Cen**[*1], **Yihang Yao**[*1], **William Han**[1], **Zuxin Liu**[2], **Ding Zhao**[1]

[1] Carnegie Mellon University, [2] Salesforce AI Research

[*] Equal contribution, {zcen, yihangya}@andrew.cmu.edu

## Abstract

Reinforcement learning (RL) has emerged as a powerful post-training technique to incentivize the reasoning ability of large language models (LLMs). However, LLMs can respond very inconsistently to RL finetuning: some show substantial performance gains, while others plateau or even degrade. To understand this divergence, we analyze the per-step influence of the RL objective and identify two key conditions for effective post-training: (1) RL-informative rollout accuracy, and (2) strong data co-influence, which quantifies how much the training data affects performance on other samples. Guided by these insights, we propose behavior injection, a task-agnostic data augmentation scheme applied prior to RL. Behavior injection enriches the supervised finetuning (SFT) data by seeding exploratory and exploitative behaviors, effectively making the model more RL-ready. We evaluate our method across two reasoning benchmarks with multiple base models. The results demonstrate that our theoretically motivated augmentation can significantly increase the performance gain from RL over the pre-RL model. Website: `https://bridge-llm-reasoning.github.io/`.

## 1 Introduction

Large language models (LLMs) have demonstrated remarkable reasoning capabilities and strong performance across a broad range of tasks [1, 2, 3], including mathematical problem solving [4, 5, 6], code generation [7, 8], and embodied decision-making [9, 10]. When guided by chain-of-thought (CoT) prompting [11], LLMs are able to generate intermediate reasoning steps, leading to more structured and interpretable outputs. Despite these advances, LLMs still struggle with complex, multi-step reasoning tasks, such as mathematical competitions [12] and real-world agentic scenarios [13, 14, 15]. A common strategy for improving performance is to scale up training corpora [16, 17, 18, 19]. However, concerns have been raised that high-quality pretraining data may soon be exhausted [20], and continued data scaling introduces significant computational overhead.

An alternative and increasingly prominent direction is to enhance LLM reasoning through reinforcement learning (RL) [21, 22], which enables reward-driven fine-tuning based on verifiable outcomes, ranging from ground-truth correctness [23, 24] to feedback from executable environments [25, 14, 15]. In traditional low-dimensional RL, performance gains are typically attributed to algorithmic improvements [26, 27] and data quality [28, 29, 30], with little attention paid to initialization since models are often trained from scratch. In contrast, the RL process for LLMs begins with a pretrained or fine-tuned model, and we argue that the quality of this initialization, particularly after supervised fine-tuning (SFT), plays a critical role in determining the effectiveness of subsequent RL. In this work, we focus on the *warm-start RL training pipeline* [23], which first fine-tunes an LLM via SFT before applying reinforcement learning. Our goal is to make language models *RL-ready*: rather than modifying RL algorithms themselves, we propose a data-centric strategy to enhance the base model's ability to benefit from RL, thereby improving training efficiency and downstream performance.

39th Conference on Neural Information Processing Systems (NeurIPS 2025).

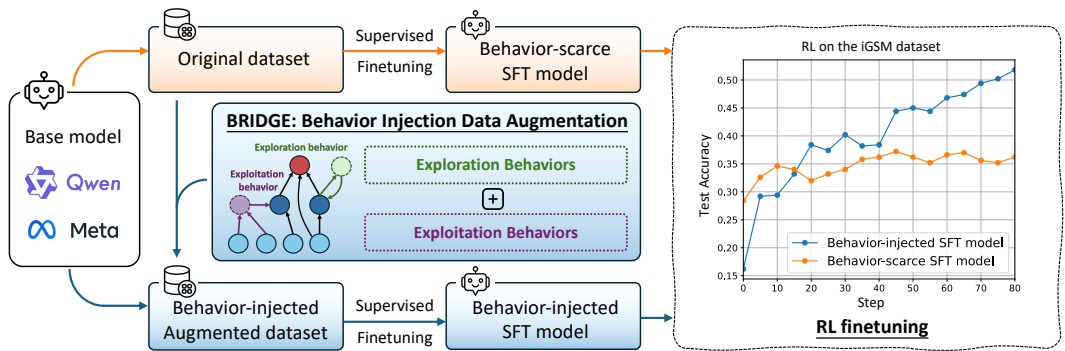

Figure 1: Overview of the BRIDGE pipeline: We augment the SFT data by introducing exploration and exploitation behaviors to prepare LLMs ready for RL finetuning.

Our key insights stem from two perspectives: (1) Analysis of the RL learning objective: we identify two critical factors that influence model improvement during RL tuning: *rollout accuracy* and the *data co-influence coefficient*, which quantifies how strongly RL training data affects generalization to the target domain. (2) Desirable behaviors for RL: while *exploration* and *exploitation* are central in low-dimensional RL [31], they are often underexplored in the context of LLM post-training. Motivated by these findings, we introduce **BRIDGE** (**B**ehavio**R** **I**njection **D**ata au**G**m**E**ntation), a data-centric augmentation strategy applied during the SFT stage. BRIDGE injects desired behaviors into the model before RL, enabling it to generate more informative trajectories during RL rollout and leading to greater final performance improvements. Our contributions are summarized as:

**1. In-depth analysis of LLM reinforcement learning.** We provide a detailed examination of the RL training process, highlighting two key factors that drive learning efficiency: rollout accuracy distribution and the data co-influence coefficient.

**2. Introduction of the BRIDGE augmentation algorithm.** We propose BRIDGE, which prepares the model for RL by explicitly injecting exploration and exploitation behaviors during SFT.

**3. Comprehensive empirical evaluation.** We evaluate BRIDGE across diverse tasks from iGSM and PromptBench. Extensive experiments and ablation studies demonstrate that BRIDGE enhances data co-influence and significantly improves performance in the RL stage.

## 2   Related Work

**RL-based post-training for LLMs.** Reinforcement learning (RL) has become a central post-training approach for aligning and extending large language models. Large-scale efforts such as OpenAI-o1 [32] and DeepSeek-R1 [33] illustrated the gains obtainable from reward optimization on general-purpose models. Since then, RL fine-tuning has been pushed into a variety of domain-specialized settings. In mathematics, verifier-guided or programmatically graded rewards help models master challenging problems [34, 35, 24, 36, 37, 38], while logic benchmarks likewise benefit from RL-driven reasoning refinement [39, 40]. Interactive agents leverage RL for textual tool use and multi-step planning [41, 42, 43, 44, 45, 46], mobile-app control [25], device manipulation [47, 48], and web navigation [49]. Additional applications include medical visual QA [50], software-engineering assistance [51], social reasoning [52], and tool-centric instruction following [53].

**Analysis of LLM finetuning.** To understand how model performance changes during finetuning, researchers study SFT learning dynamics of LLM [54] in terms of data influence [55, 56] or likelihood analysis [57, 58]. In the context of RL finetuning, OpenAI o1 [32] showed that RL significantly improve reasoning by encouraging the generation of longer CoT. Subsequent studies [59, 60, 61, 62] validate this effect and show that RL enables inference-time scaling by favoring more expressive reasoning traces. This finding is aligned with theoretical analyses that characterize the expressivity of CoT [63, 64, 65, 66]. Furthermore, researchers [67, 68, 69] observe that RL fine-tuning often amplifies behaviors already accessible in the base model rather than introducing entirely new ones, which are crucial cognitive operations to performance growth in RL [70]. Distinct from these works, we investigate the tuning dynamics by RL learning objective, and we identify behaviors from the perspective of exploration and exploitation to prepare models for RL tuning.

**Data-centric approaches for LLMs.** A complementary line of work improves language models not by changing the training algorithm but by enriching the data they see. In supervised fine-tuning (SFT), targeted augmentation has proved especially fruitful: synthetic derivations, curriculum sampling, or structure-preserving rewrites boost mathematical reasoning [16, 71, 72, 73, 74, 4] and extend to code generation domains [75]. Beyond manual augmentation, agentic pipelines automate data acquisition, where LLM-based agents write queries, run tools, and self-filter their outputs, yielding high-quality instruction data at a large scale [76, 19, 77, 78, 79, 80]. For RL settings, several studies curate *seed* datasets whose answers can be programmatically verified, seeding reward-based training with reliable trajectories [81, 14]. Others examine what kinds of pre-RL corpora make a base model more amenable to later policy optimization, highlighting the importance of coverage, diversity, and error profiles [62, 70]. Our work aligns with this data-centric perspective but focuses on *behavior-level* augmentations that explicitly *make models RL-ready*, rather than merely enlarging or filtering the SFT corpus.

## 3 Method

In this section, we first investigate key factors that affect the model performance growth in the RL stage, and then introduce our method based on the analysis to make the pre-RL model "RL-ready", i.e., being able to boost the performance after the RL stage.

### 3.1 Preliminaries

Following the previous pipeline [23], we apply SFT to the base models, followed by an RL stage. **Supervised finetuning (SFT)**. SFT is a common practice to initialize LLM finetuning by a demonstration dataset $\mathcal{D}_{\text{SFT}}$, which trains the policy $\pi_\theta$ by minimizing the negative log-likelihood:

$$\min_\theta \mathcal{L}_{\text{SFT}}(\mathcal{D}; \theta) = -\mathbb{E}_{(\mathbf{q},\mathbf{a}) \sim \mathcal{D}_{\text{SFT}}} \left[ \sum_{t=1}^{T} \log \pi_\theta \left( a_t \mid \mathbf{q}, \mathbf{a}^{(<t)} \right) \right], \tag{1}$$

where $\mathbf{q} = [q_1, \ldots, q_L], \mathbf{a} = [a_1, \ldots, a_T]$ are demonstration query and answer from SFT dataset. For simplicity, we use SFT model to refer to the model after SFT training.

**RL finetuning**. The objective of RL for LLM is to maximize the expectation of reward on a query set $Q$:

$$\max_\theta \mathcal{J}_{\text{RL}}(Q; \theta) = \mathbb{E}_{\mathbf{q} \sim Q, \mathbf{o} \sim \pi_\theta(\cdot|\mathbf{q})} \left[ \sum_{t=1}^{T} \gamma^t r \left( \mathbf{q}, \mathbf{o}^{(\leq t)} \right) \right], \tag{2}$$

where $\mathbf{q} = [q_1, \ldots, q_L], \mathbf{o} = [o_1, \ldots, o_T]$ indicate the query and output respectively, $r(\cdot, \cdot)$ is a reward function, and $\gamma$ is the discount factor. We mainly consider the rule-based outcome reward, i.e., a binary reward on the last token of output:

$$r(\mathbf{q}, \mathbf{o}) = \mathbf{1}(y(\mathbf{o}) = y_{\text{gold}}),$$

where $y(o)$ is the final answer of the output and $y_{\text{gold}}$ is the ground-truth answer of the query. In this paper, we adopt GRPO [23] as the RL algorithm, which samples $N$ outputs $\{\mathbf{o}_i\}_{i=1}^{N}$ for each query and optimizes a surrogate objective:

$$\mathcal{J}(Q; \theta) = \mathbb{E}_{\mathbf{q} \sim Q, \{\mathbf{o}_i\} \sim \pi_\theta(\cdot|\mathbf{q})}$$

$$\frac{1}{N} \sum_{i=1}^{N} \left[ \min \left\{ \frac{\pi_\theta(\mathbf{o}_i|\mathbf{q})}{\pi_{\text{old}}(\mathbf{o}_i|\mathbf{q})} A_i, \text{clip} \left( \frac{\pi_\theta(\mathbf{o}_i|\mathbf{q})}{\pi_{\text{old}}(\mathbf{o}_i|\mathbf{q})}, 1 - \epsilon, 1 + \epsilon \right) A_i \right\} - \beta D_{\text{KL}}(\pi_\theta \| \pi_{\text{ref}}) \right], \tag{3}$$

where $\epsilon$ is the clip ratio, $\beta$ is the KL regularization coefficient, and $A_i$ is the advantage computed by normalizing the reward in each group $A_i \doteq (r_i - \text{mean}(\{r_i\}_{i=1}^{N}))/\text{std}(\{r_i\}_{i=1}^{N})$.

### 3.2 Training Per-step Influence in RL Finetuning

To understand why LLMs respond to RL divergently, we leverage **per-step influence** [55, 82] to answer the question *how the model performance changes after one RL training step.*

Consider a language model policy $\pi_\theta$, suppose we update it on one query-output group $(\mathbf{q}, \{\mathbf{o}_i\}_{i=1}^N)$ with learning rate $\eta$, then the parameter update is $\Delta\theta = \eta\nabla_\theta\mathcal{J}(\mathbf{q};\theta)$ since the RL objective is to maximize $\mathcal{J}$. According to Taylor expansion, the caused model performance change on other query $\mathbf{q}'$ can be written as

$$\Delta\mathcal{J}(\mathbf{q}';\theta) = \mathcal{J}(\mathbf{q}';\theta + \Delta\theta) - \Delta\mathcal{J}(\mathbf{q}';\theta) = \langle\nabla_\theta\mathcal{J}(\mathbf{q}';\theta), \Delta\theta\rangle + \mathcal{O}(\eta^2).$$

With a sufficiently small learning rate, the per-step influence of $\mathbf{q}$ on the model performance is

$$\Delta\mathcal{J}(Q;\theta) \approx \eta\langle\nabla_\theta\mathcal{J}(Q;\theta), \nabla_\theta\mathcal{J}(\mathbf{q};\theta)\rangle. \tag{4}$$

Then we can derive the per-step influence for the GRPO objective as follows.

**Proposition 3.1.** *Suppose there are $n$ correct outputs in the sampled group with size $N$, denote the correct and incorrect outputs as $\{\mathbf{o}_{i+}\}_{i=1}^n$ and $\{\mathbf{o}_{j-}\}_{j=1}^{N-n}$ respectively. When RL training on $\mathbf{q}$ is strictly on policy, with a sufficiently small $\beta$, the per-step influence of $\mathbf{q}$ on the model performance is*

$$\Delta\mathcal{J}(Q;\theta) = \eta\mathbb{E}_{\mathbf{q}'\sim Q, \mathbf{o}'\sim\pi_\theta(\cdot|\mathbf{q}')}\sqrt{\alpha(1-\alpha)}$$
$$A(\mathbf{q}',\mathbf{o}')\cdot\left[\frac{1}{n}\sum_{i=1}^n\mathcal{K}_\theta[(\mathbf{q}',\mathbf{o}'),(\mathbf{q},\mathbf{o}_{i+})] - \frac{1}{N-n}\sum_{j=1}^{N-n}\mathcal{K}_\theta[(\mathbf{q}',\mathbf{o}'),(\mathbf{q},\mathbf{o}_{j-})]\right], \tag{5}$$

*where $\alpha = n/N$ indicates the accuracy rate for the rollout samples, $\mathcal{K}_\theta((\mathbf{q}',\mathbf{o}'),(\mathbf{q},\mathbf{o})) = \langle\nabla_\theta\log\pi_\theta(\mathbf{o}'|\mathbf{q}'), \nabla_\theta\log\pi_\theta(\mathbf{o}|\mathbf{q})\rangle$ indicates the influence between the log-likelihood of query-output samples $(\mathbf{q}',\mathbf{o}')$ and $(\mathbf{q},\mathbf{o})$.*

The proof and discussion are in Appendix A.1. The derivation is mainly based on two assumptions, the on-policy training and a small KL coefficient $\beta$, both of which hold in practical implementation. Note that this proposition can also be extended to other RL algorithms by replacing the advantage function. More discussion is provided in Appendix A.2.

This proposition shows that the per-step influence is mainly determined by two factors: (1) the accuracy rate of sampled output, $\alpha$, and (2) the co-influence coefficient $\mathcal{K}_\theta$ between training data $(\mathbf{q},\mathbf{o})$ and target data $(\mathbf{q}',\mathbf{o}')$. For the first factor, $0\%$ or $100\%$ rollout accuracy makes the coefficient $\sqrt{\alpha(1-\alpha)} = 0$, leading to $\Delta\mathcal{J} = 0$, while a medium accuracy can amplify the influence, which aligns with previous theoretical results on reward variance [83] and practical observation [84]. The second factor, expressed through $\mathcal{K}_\theta[\cdot,\cdot]$, quantifies how strongly each training sample affects performance change on the target domain. Take a correct target sample (i.e., $A(\mathbf{q}',\mathbf{o}') > 0$) for example, we expect a large positive $\mathcal{K}_\theta$ when the update draws on correct samples $(\mathbf{q},\mathbf{o}_{i+})$, enabling the language model to learn more from this successful path and gain greater improvement in the RL step. Conversely, when the update uses incorrect samples $(\mathbf{q},\mathbf{o}_{j-})$, we expect $\mathcal{K}_\theta$ for the same correct target sample to be small or negative, allowing the model to learn from the failure path. We defer the detailed explanation of the validation method to examine this factor to section 4.3.

> **Section 3.2 takeaway**
>
> There are two key strategies to enhance performance growth during the RL stage of LLMs:
> - Altering the accuracy distribution to increase the information coefficient $\sqrt{\alpha(1-\alpha)}$.
> - Shaping the co-influence coefficient of model, governed by $\mathcal{K}_\theta$, to improve knowledge acquisition from both successful and failed rollout experiences.

### 3.3 BRIDGE: Behavior Injection Data Augmentation

Based on the above analysis, the next question is how to improve performance gains during RL in practical implementations. A straightforward approach is to perform rejection sampling on queries from low-information accuracy regions, i.e., those with excessively high or low $\alpha$. However, this method has three key drawbacks: (1) Increased computational cost: it requires pre-sampling to identify and discard low-information samples; (2) Distortion of the query distribution: it induces distribution shift issues in optimization; and (3) Limited impact on data co-influence: this approach does little to affect the data co-influence factors involving $\mathcal{K}_\theta$.

Rather than directly reshaping the RL training distribution, we focus on the characteristics of the base model, which fundamentally influence both the accuracy distribution and the co-influence

structure. In particular, we target two essential capabilities of RL agents: *exploration* and *exploitation*. Exploration encourages the model to traverse a broader range of observations and actions, helping it avoid suboptimal local modes. Exploitation enhances the model's ability to leverage its current knowledge for effective decision-making, which can in turn increase the co-influence of training data during RL. By injecting these two behaviors into the base LLMs, we promote more effective learning dynamics and greater performance gains in RL training.

We implement behavior injection through our proposed method, BRIDGE (**B**ehavio**R** **I**njection **D**ata au**G**m**E**ntation), as illustrated in Figure 1. Starting from a vanilla chain-of-thought (CoT) dataset for instruction tuning, we augment the dataset by injecting exploration and exploitation behaviors. We then perform supervised fine-tuning (SFT) on the augmented dataset, followed by reinforcement fine-tuning (RL).

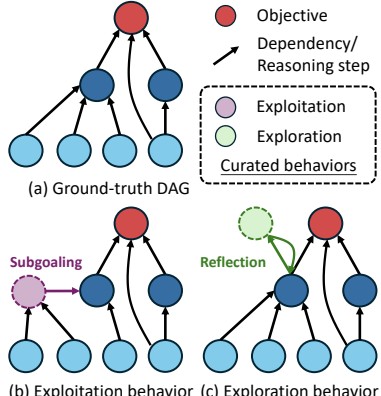

(a) Ground-truth DAG

(b) Exploitation behavior   (c) Exploration behavior

Figure 2: DAG representation of behaviors.

---

**Algorithm 1:** BRIDGE

**Input:** Vanilla QA pairs $(q, a)$, injection prob $p$
**Output:** Injected QA pairs $(q, a')$
1: Extract DAG $\mathcal{G} = (V, E)$ from $(q, a)$.
2: # Construct exploration behaviors.
3: Obtain a locked node $n_l$ ahead.
4: $b_1$ = "Let's solve $[n_l]$, ..., wait, $[n_l]$ seems to be not solvable yet, let's get back."
5: # Construct exploitation behaviors.
6: Aggregate all the information `info` to solve an unlocked but not solved node $n_u$
7: $b_2$ = "Let's solve $[n_u]$, ..., since $[\text{info}]$, $[n_u]$ =[subgoal computation]=$[n_u.\text{value}]$"
8: $a' \leftarrow a + b_1 + b_2$; # Inject with prob $p$.
9: **Return:** Injected QA pair $(q, a')$

---

To further ground our approach in a practical implementation, we adopt a directed acyclic graph (DAG) representation for reasoning tasks [85], as shown in Figure 2. Formally, a reasoning task is represented as $\mathcal{G} = (V, E)$, where $V$ is the set of nodes and $E \subseteq V \times V$ denotes the directed edges that capture dependencies and relationships between nodes. Given initial information and inter-node dependencies, the goal is to compute intermediate node results step by step and ultimately derive the final answer at the target node. The ground-truth reasoning chain corresponds to a topological sort of the DAG. Notably, many reasoning and agentic tasks, such as math reasoning [86, 87, 88] and logical reasoning [86, 73], naturally conform to this graph-based representation [77].

Our BRIDGE approach is detailed in Algorithm 1. We first extract the DAG representation from the original query and its corresponding CoT response. Note that such extraction can be achieved by string matching for structured CoT or an oracle LLM for unstructured CoT. More discussion is in the Appendix B.2. After constructing the behaviors, we generate *exploration behaviors* by attempting to solve a locked (i.e., not yet solvable) node ahead of time, followed by reflection. For *exploitation behaviors*, we actively aggregate available information to solve an unlocked node (i.e., solvable but not yet solved) or compute a sub-goal instead of directly reaching the node value, reinforcing the known reasoning path. After generating these behaviors, we inject them into the vanilla CoT corpus, resulting in an augmented dataset.

## 4 Experiment

### 4.1 Experiment settings

**Tasks**. To evaluate the performance growth during RL finetuning, we conduct experiments on two benchmarks: 1) iGSM [74], a grade-school math problem benchmark involving math and common sense reasoning tasks; 2) PromptBench [86], a benchmark involving arithematic and logical reasoning tasks. Note that the difficulty of the tasks in both benchmarks are controllable: iGSM controls the difficulty of the query by the *number of operations* needed to reach the final answer while PromptBench controls it by *reasoning depth* and *number of redundant premises* (i.e., the redundant edges in DAG) in query. We adopt the strict match accuracy as the evaluation metrics for both tasks.

In practice, we apply slight modifications on iGSM to improve the finetuning stability, e.g., removing the modulo operation because we observe that the base models are unable to calculate the modulo with high accuracy and it cannot be significantly enhanced by training on a small SFT dataset [74]. See Appendix B.1 for more details of the tasks.

We select these two benchmarks for analysis because we want to test the problem-solving abilities of LLMs rather than the knowledge storage. While other tasks where domain knowledge can confound the evaluation of reasoning capabilities, the purely synthetic data in iGSM and PromptBench avoid the data contamination issue fundamentally [74, 89]. Although built upon relatively small semantics domains, these tasks require multi-dimensional abilities of LLM such as basic concept understanding, search and planning, and basic arithmetic calculation, which mirrors the key aspects of problem-solving in general reasoning tasks.

**Injected behaviors**. We adopt two exploitative behaviors, *subgoal computation* and *information analysis*, and one exploratory behavior, *reflection*, respectively. Specifically, *subgoal computation* refers to computing intermediate results for complex equations; *information analysis* involves aggregating relevant information when deriving the equation for a node; *reflection* means actively attempting to solve an unlocked node and then revisiting back. We add the *subgoal computation* to each step of CoT but inject the *analysis* and *reflection* with probability $p = 0.1$. Detailed examples of behaviors in iGSM and PromptBench tasks are illustrated in Appendix B.3.

**Baselines**. We compare BRIDGE with several data augmentation baselines: 1) `Vanilla`, which uses original SFT data without augmentation; 2) premise permutation augmentation (`PP-Aug`) [90, 73], which augments SFT data by randomly shuffling the premises in the query to improve the reasoning consistency; 3) reasoning chain augmentation (`RC-Aug`) [4], which is implemented by generating new answers with different topological orders for the same query from SFT dataset.

**Other experiment settings**. We use two families of base models, Qwen-2.5 [91] and Llama-3.2 [92], to validate the effectiveness of our methods. As there is a large domain gap between the pretraining corpus and evaluation tasks, we first train base LLMs on a SFT dataset to expose them to the query set and demonstration answers. Then we use GRPO with the same settings to finetune the SFT models for different augmentation methods. More training details are attached in Appendix B.4.

## 4.2 Main results

We present the performance comparison on iGSM and PromptBench in Table 1 and 2 respectively.

For the iGSM task, we use data with $15 \sim 20$ operations for finetuning. In SFT stage, the vanilla dataset consists of 2000 data while `PP-Aug` and `RC-Aug` augments dataset size to 8000. BRIDGE also uses 2000 data for SFT but augments them by injecting behaviors. We train each model on corresponding dataset for 5 epochs. In the RL stage, we train on the data with the same difficulty. To avoid overfitting (e.g., the memorization in SFT [93, 39]) and evaluate the generalizability of the finetuned models, we test their performance on two problem sets separately: 1) in-distribution

Table 1: The evaluation results (%) on the iGSM task. We train SFT models for 5 epochs and finetune them with RL for 100 (Qwen 3B) or 200 steps (Qwen 1.5B and Llama 1B). We compare SFT and RL models on both in-distribution and out-of-distribution problem sets. $\Delta$ denotes the performance improvement by RL over SFT.

| Base | | Vanilla | | PP-Aug | | RC-Aug | | BRIDGE (Ours) | |
|---|---|---|---|---|---|---|---|---|---|
| | | In-Dist | OOD | In-Dist | OOD | In-Dist | OOD | In-Dist | OOD |
| Qwen-1.5B | SFT | 40.2 | 29.0 | 46.6 | 33.4 | 33.2 | 27.4 | 44.8 | 34.2 |
| | RL | 46.2 | 36.4 | 60.4 | 48.0 | 46.2 | 39.6 | **91.4** | **83.0** |
| | $\Delta$ | 6.0 | 7.4 | 13.8 | 14.6 | 13.0 | 12.2 | **46.6** | **48.8** |
| Qwen-3B | SFT | 38.0 | 28.4 | 38.0 | 19.8 | 45.6 | 32.8 | 59.2 | 45.8 |
| | RL | 57.2 | 47.4 | 62.4 | 52.0 | 60.6 | 45.6 | **89.6** | **83.6** |
| | $\Delta$ | 19.2 | 19.0 | 24.4 | 32.2 | 15.0 | 12.8 | **30.4** | **37.8** |
| Llama-1B | SFT | 29.6 | 19.0 | 32.4 | 23.2 | 31.0 | 21.4 | 40.4 | 26.6 |
| | RL | 31.0 | 24.0 | 39.0 | 26.4 | 33.6 | 23.4 | **64.6** | **44.8** |
| | $\Delta$ | 1.4 | 5.0 | 6.6 | 3.2 | 2.6 | 2.0 | **24.2** | **18.2** |

Table 2: The evaluation results (%) on the PromptBench arithmetic reasoning task. We train SFT models for 2 epochs and finetune them with RL for 100 steps. We compare performances on both in-distribution and OOD problem sets. $\Delta$ denotes the performance improvement by RL over SFT. **Bold** means the best performance.

| Base | | Vanilla | | PP-Aug | | RC-Aug | | BRIDGE (Ours) | |
|---|---|---|---|---|---|---|---|---|---|
| | | In-Dist | OOD | In-Dist | OOD | In-Dist | OOD | In-Dist | OOD |
| Qwen-1.5B | SFT | 13.6 | 9.4 | 16.8 | 8.8 | 18.6 | 13.0 | 2.2 | 0.8 |
| | RL | 37.8 | 29.0 | 41.2 | 31.2 | 34.2 | 21.6 | **55.2** | **41.0** |
| | $\Delta$ | 24.2 | 19.6 | 24.4 | 22.4 | 15.6 | 8.6 | **53.0** | **40.2** |
| Qwen-3B | SFT | 31.2 | 18.0 | 40.6 | 29.4 | 35.2 | 19.4 | 44.4 | 31.0 |
| | RL | 50.0 | 34.6 | 66.0 | 50.8 | 64.0 | 43.4 | **85.8** | **70.0** |
| | $\Delta$ | 18.8 | 16.6 | 25.4 | 21.4 | 28.8 | 24.0 | **41.4** | **39.0** |
| Llama-1B | SFT | 12.6 | 5.4 | 11.2 | 8.8 | 9.2 | 4.4 | 6.0 | 3.6 |
| | RL | 27.8 | 18.8 | 28.6 | 19.0 | 25.2 | 17.4 | **47.6** | **39.8** |
| | $\Delta$ | 15.2 | 13.4 | 17.4 | 10.2 | 16.0 | 13.0 | **41.6** | **36.2** |

(In-Dist) set with operation number $= 20$ and 2) out-of-distribution (OOD) set with operation number $= 25$. Each set consists of 500 problems. We use greedy sampling and compute the accuracy when evaluating on the test sets.

For the PromptBench task, we use data with reasoning depth $= 4$ and 5 for SFT and RL training respectively, both of which have $0 \sim 8$ redundant premises. This simulates the real-world settings that we have labeled CoT answers for easy tasks and only have verifiers for harder tasks where the demonstration annotation is absent. In SFT stage, we train the models on 5000 data (`PP-Aug` and `RC-Aug` augment it to 10000) for 2 epochs. In RL stage, we train the models for 100 RL steps. We also test the performances on 1) in-distribution set with reasoning depth $= 5$ and $0 \sim 8$ redundancy and 2) out-of-distribution set with reasoning depth $= 5$ and $20 \sim 22$ redundancy separately.

In iGSM task, most augmentation methods with different base models increase the SFT accuracy on in-distribution set with models while the improvement on OOD set is less prominent. For RL, BRIDGE achieves both the highest performance growth and best accuracy on in-distribution and OOD sets. In PromptBench task, although BRIDGE has relatively lower pre-RL accuracy, it obtains significantly more remarkable performance growth than baselines during RL and thus achieves the best final score on both in-distribution and OOD sets. The training curves and more training results are in Appendix B.5 where we also present a result of comparison between BRIDGE and accuracy-based rejection sampling.

### 4.3 Rollout accuracy and Co-influence of Different SFT models

To study how different augmentations lead to divergent RL performances, we test the rollout accuracy $\alpha$ distribution and the finetuning per-step influence $\Delta \mathcal{J}$ of the SFT models on iGSM task. The results are illustrated in Fig 3.

Specifically, we use the SFT model to rollout 8 samples on a iGSM test set consisting of 2000 data with the same difficulty (operation number $= 15 \sim 20$) as the RL training set. We compute the advantages of each answer (by normalizing the reward within the group) and the sampling accuracy distribution of each model based on the rollout results. When measuring the data co-influence $\mathcal{K}_\theta$, the directly computation requires taking the inner product between two gradients with the same size as the model parameters, which is intractable to scale to a large dataset. Instead, we adopt a low-rank approximation [82] that first estimates the LoRA [94] gradients of SFT model and then applies random projection to further reduce the dimensionality. Based on the advantages and the approximated data co-influence $\tilde{\mathcal{K}}$, we can compute the approximated per-step influence of one query-output group on other data $\Delta \tilde{\mathcal{J}}$ by Eq.(5). More implementation details are in Appendix B.7.

We present the sampling accuracy, approximated per-step influence and training curves of different SFT models in Figure 3. In both settings, BRIDGE has the fastest reward growth during the training, which can be explained in terms of both sampling accuracy and data co-influence in finetuning. All of three augmentations increase the accuracy of SFT model, but `RC-Aug` achieves it mainly by more fully correct samples (acc=1). On the contrary, BRIDGE has the largest ratio of samples with

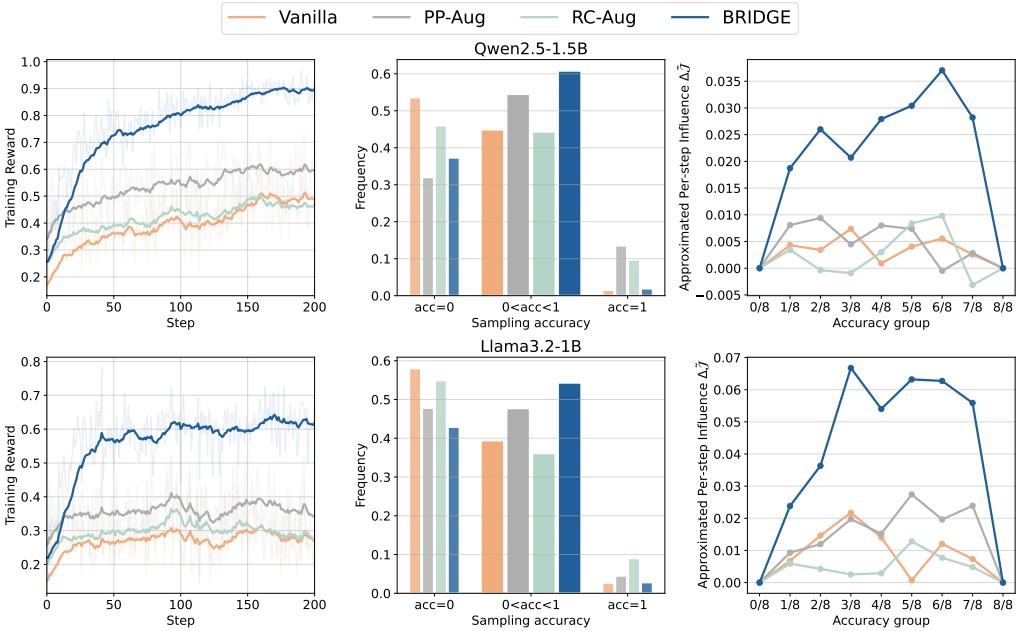

Figure 3: **Left:** Training curve, **Middle:** SFT model rollout accuracy distribution, **Right:**. Per-step influence visualization. The top and bottom plots correspond to the results of Qwen2.5-1B and Llama3.2-1B respectively. We group the approximated per-step influence for samples with different accuracy in right plots, where the influences of samples with all correct or all wrong answers are 0.

medium accuracy ($0 <$acc$< 1$) that contribute to model improving during RL training, even larger than PP-Aug which has better performance at the beginning of RL. Meanwhile, the right plots clearly show the discrepancy among different methods: while PP-Aug and RC-Aug models have similar per-step influence with vanilla SFT model, our method significantly improves it by proper behavior injection into the SFT data, and thus better prepares the model for RL.

> **Section 4.2 and 4.3 takeaway**
>
> BRIDGE demonstrates superior capability to boost RL performance (Table 1, 2), with:
> - Injected behaviors improve the per-step influence $\Delta \mathcal{J}(Q; \theta)$ in Eq.(5). (Figure 3 **Right**).
> - Injected behaviors shape the accuracy distribution of rollout samples (Figure 3 **Middle**).

## 4.4 Ablations of Injected Behaviors

As we add various behaviors to the demonstration dataset to prepare the model for RL, it remains unclear how each behavior contributes to the final improvement. Therefore, we run an ablation by adding only one behavior and present the results in Figure 4.

For Qwen2.5-1.5B models, the *subgoal computation* increases the SFT model accuracy and all three behaviors improve the performance gain in RL, consistent with their per-step influences. For Llama3.2-1B, the models with single behavior still outperform the baseline but the performance gain in RL is marginal. Meanwhile, we observe that there is a large accuracy gap between the model trained by BRIDGE and model with any single behavior after RL finetuning, suggesting the necessity of behavior combination for the Llama model.

We also present ablation studies on the injection probability of behaviors for the iGSM task in Figure 5. Comparing $p = 0$ with $p > 0$, we observe that the injected *reflection* and *analysis* behaviors consistently enhances the final performance. Furthermore, performance differences across various non-zero injection probabilities are nuanced. This suggests that the exact behavior ratio is not critical, and LLMs can adaptively learn to utilize these behaviors during RL as long as the behavior is injected into the model.

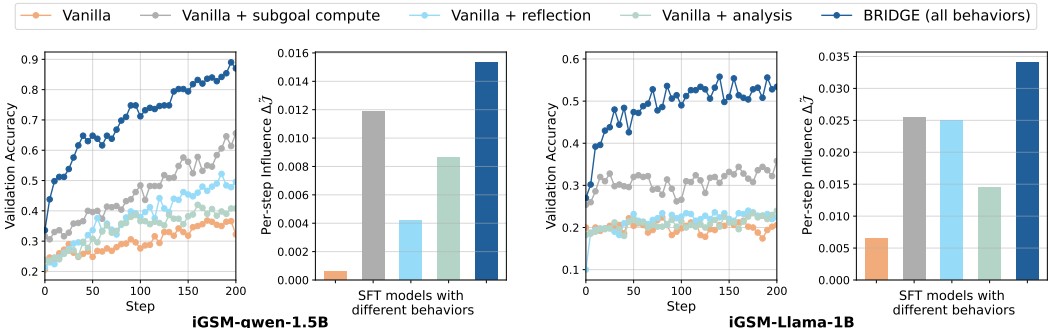

Figure 4: The ablation of models with different behaviors in iGSM task. We present the validation accuracy curves as the RL finetuning performances, where the validation set consists of 500 queries with $21 \sim 25$ operations. We also compare the average per-step influence of the SFT models with different behaviors.

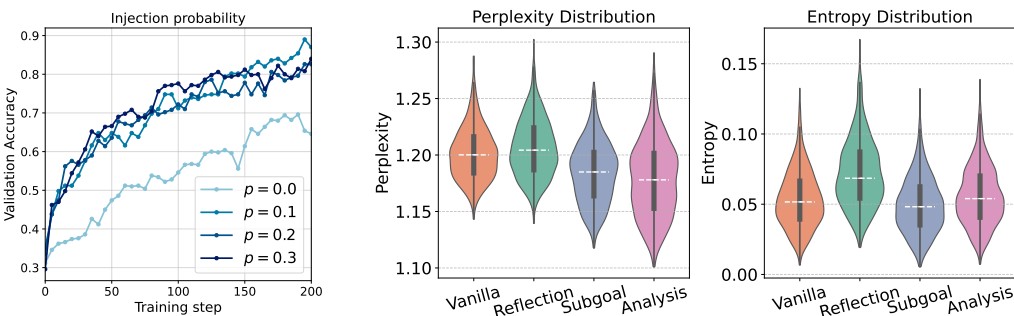

Figure 5: The ablations on behavior injection probability.

Figure 6: The perplexity and entropy of the SFT models.

## 4.5 Exploration and Exploitation Effects of Injected Behaviors

In this section, we study the effects of behaviors on exploration and exploitation. We leverage policy entropy and perplexity as the corresponding metrics: we first train the model on datasets with the same size for the same training steps. Then we rollout the SFT models with temperature $= 1$ (the same as rollout in RL finetuning) and compute the policy entropy on model's generation; we also use the log-likelihood on corresponding SFT dataset to compute the perplexity. The results are presented in Figure 6.

While all above injected behaviors contribute to the preparation of RL finetuning, their effects on RL work from different aspects: *subgoal computation* and *analysis* reduce the perplexity on demonstration data and enable the models to generate the ground-truth tokens with higher probability, suggesting these behaviors facilitate the exploitation of LLM. Meanwhile, the *reflection* behavior leads to a larger policy entropy in generation, validating its effectiveness in encouraging LLMs to explore. In BRIDGE, we inject both types of behaviors to LLMs to enhance both exploration and exploitation of LLMs, making them significantly more "RL-ready" compared to the vanilla models.

> **Section 4.4 and 4.5 takeaway**
>
> The injected behaviors can either improve exploration or exploitation capability, specifically,
> • Behavior injection is not sensitive to the behavior density, as long as behaviors are successfully injected into the pre-RL model. (Figure 5)
> • Both exploration and exploitation behaviors injected through BRIDGE are effective to improving per-step influence $\Delta \mathcal{J}(Q; \theta)$ in Eq.(5), thus benefiting RL. (Figure 4)
> • Reflection improves exploration while subgoal computation and analysis improve exploitation, as examined by model entropy and perplexity. (Figure 6)

# 5    Conclusion

In this paper, we investigate why different language models manifest divergent performances during RL finetuning by analyzing the per-step influence of data for GRPO algorithm. We then propose to inject behaviors, which are RL-favorable in terms of exploration and exploitation, into the SFT dataset to prepare LLMs for RL finetuning. Through various experiments across different benchmarks, we demonstrate that our method effectively makes the models RL-ready and significantly outperforms other augmentation baselines.

One limitation of BRIDGE is that the evaluation is focused on the math and common-sense reasoning tasks. The future work involves extending our method to broader agentic domains with a larger dataset size. Meanwhile, we believe the analysis tools (e.g., per-step influence computation) can inspire future work on data curation and behavior discovery in the community. One potential negative social impact of this paper is the misuse of our method, and unsafe behavior injection can lead to harmful consequences.

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

# A  Proofs and Discussions of Theoretical Results

## A.1  The Proof and Discussion of Proposition 3.1

The proof is as follows:

*Proof.* We first compute the advantage in one group. For query $\mathbf{q}$ with $n$ correct outputs and $N - n$ wrong outputs ($0 < n < N$), the mean and standard deviation of rewards are $n/N$ and $\sqrt{n(N-n)}/N$ respectively. Accordingly, the advantages of positive and negative samples are

$$
\begin{cases}
A(\mathbf{q}, \mathbf{o}_+) & = \frac{1-n/N}{\sqrt{n(N-n)}/N} = \sqrt{\frac{N-n}{n}} \\
A(\mathbf{q}, \mathbf{o}_-) & = \frac{0-n/N}{\sqrt{n(N-n)}/N} = -\sqrt{\frac{n}{N-n}}
\end{cases}, \tag{6}
$$

Note that the advantage will be all 0 if $n = 0$ or $n = N$.

Then we consider the GRPO objective in Eq.(3), when the RL training is on policy, the importance ratio $\pi_\theta(\mathbf{o}|\mathbf{q})/\pi_{\text{old}}(\mathbf{o}|\mathbf{q}) = 1$ and thus we can remove the corresponding clip terms. Meanwhile, we ignore the KL divergence regularization when the coefficient $\beta$ is small. Consequently, the gradient of simplified GRPO objective on query $\mathbf{q}$ is

$$
\nabla_\theta \mathcal{J}(\mathbf{q}; \theta) \tag{7}
$$

$$
= \mathbb{E}_{\{\mathbf{o}_i\} \sim \pi_\theta} \frac{1}{N} \sum_{i=1}^{N} \left[ \frac{\nabla_\theta \pi_\theta(\mathbf{o}_i|\mathbf{q})}{\pi_{\text{old}}(\mathbf{o}_i|\mathbf{q})} A_i \right] \tag{8}
$$

$$
= \mathbb{E}_{\{\mathbf{o}_i\} \sim \pi_\theta} \frac{1}{N} \sum_{i=1}^{N} \left[ \frac{\pi_\theta(\mathbf{o}_i|\mathbf{q})}{\pi_{\text{old}}(\mathbf{o}_i|\mathbf{q})} \nabla_\theta \log \pi_\theta(\mathbf{o}_i|\mathbf{q}) A_i \right] \tag{9}
$$

$$
= \mathbb{E}_{\{\mathbf{o}_i\} \sim \pi_\theta} \frac{1}{N} \left[ \sum_{i=1}^{n} \nabla_\theta \log \pi_\theta(\mathbf{o}_{i+}|\mathbf{q}) A_{i+} + \sum_{j=1}^{N-n} \nabla_\theta \log \pi_\theta(\mathbf{o}_{j-}|\mathbf{q}) A_{j-} \right] \tag{10}
$$

$$
= \mathbb{E}_{\{\mathbf{o}_i\} \sim \pi_\theta} \frac{\sqrt{n(N-n)}}{N} \left[ \frac{1}{n} \sum_{i=1}^{n} \nabla_\theta \log \pi_\theta(\mathbf{o}_{i+}|\mathbf{q}) - \frac{1}{N-n} \sum_{j=1}^{N-n} \nabla_\theta \log \pi_\theta(\mathbf{o}_{j-}|\mathbf{q}) \right] \tag{11}
$$

$$
= \mathbb{E}_{\{\mathbf{o}_i\} \sim \pi_\theta} \sqrt{\alpha(1-\alpha)} \left[ \frac{1}{n} \sum_{i=1}^{n} \nabla_\theta \log \pi_\theta(\mathbf{o}_{i+}|\mathbf{q}) - \frac{1}{N-n} \sum_{j=1}^{N-n} \nabla_\theta \log \pi_\theta(\mathbf{o}_{j-}|\mathbf{q}) \right] \tag{12}
$$

where $\alpha = n/N$ and we use the policy gradient $\nabla_\theta \pi_\theta(\mathbf{q}|\mathbf{o}) = \pi_\theta(\mathbf{q}|\mathbf{o}) \nabla_\theta \log \pi_\theta(\mathbf{q}|\mathbf{o})$ in derivation.

Similarly, if we do not expand the outputs to correct and incorrect ones in one group, the gradient of GRPO objective on the whole query set is

$$
\nabla_\theta \mathcal{J}(Q; \theta) = \mathbb{E}_{\mathbf{q}' \sim Q, \{\mathbf{o}'\} \sim \pi_\theta} \left[ \nabla_\theta \log \pi_\theta(\mathbf{o}'|\mathbf{q}') A_i \right] \tag{13}
$$

Therefore, we can obtain the per-step influence by the inner product of $\nabla_\theta \mathcal{J}(\mathbf{q}; \theta)$ and $\nabla_\theta \mathcal{J}(Q; \theta)$. $\qquad \square$

*Remark* A.1 (Discussions on the assumptions). Our derivation of per-step influence relies on the on-policy training and small KL regularization coefficient assumptions, both of which approximately hold in practical implementation. The on policy training holds when one rollout step corresponds to one optimization step (i.e., one gradient descent step), which is consistent with our implementation (see Appendix B.4), other framework such as open-R1 [95], or DeepSeekMath [23] where each generation is trained only for once. Moreover, the clip term further helps reduce the gap between $\pi_\theta$ and $\pi_{\text{old}}$ even when strictly on-policy training is not guaranteed. For the KL regularization, we observe that most implementations set a very small coefficient $\beta$ (e.g., 0.01 or smaller), consistent with our assumption. Overall, the per-step influence is mainly dominated by the data co-influence as derived in Proposition 3.1.

*Remark* A.2 (Relation of data co-influence to neural tangent kernel (NTK)). The data co-influence $\mathcal{K}_\theta$ is also related to NTK [96], a kernel used to describe how neural network evolves during training via gradient descent. Denote $\chi \doteq (\mathbf{q}, \mathbf{o})$ as the query-output pair, If we regard the language model as a neural network with $\chi$ as input and the logit $\mathbf{z}$ on the whole vocabulary (with size $|V|$) of each token as output, we have

$$\nabla_\theta \log \pi_\theta(\mathbf{o}|\mathbf{q}) = \frac{1}{T} \sum_{t=1}^{T} (\pi_\theta^{(t)}(\cdot|\chi) - \mathbf{e}(o_t))^\intercal \nabla_\theta \mathbf{z}_t(\cdot|\chi) \tag{14}$$

where $T$ is the length of output $\mathbf{o}$, $o_t$ denotes the $t$-th token of the output, $\mathbf{z}_t \in \mathbb{R}^{|V|}$ is the logits of $t$-th token. Here $\pi_\theta^{(t)}(\cdot|\chi), \mathbf{e}(o_t) \in \mathbb{R}^{|V|}$. $\pi_\theta^{(t)}(\cdot|\chi)$ is the output distribution of $t$-th token and $\mathbf{e}(o_t)$ means the one-hot vector. Note that the result is divided by $T$ because there is nothing. Although we feed the model with the full answer sequence $\mathbf{o}$ in $\chi$, the tokens in later position will not contribute to computing the output distribution of each token because of the existence of causal mask, which is natively integrated in auto-regressive language model.

Then the data co-influence can be decomposed as

$$\mathcal{K}_\theta[\chi, \chi'] = \langle \nabla_\theta \log \pi_\theta(\mathbf{o}|\mathbf{q}), \nabla_\theta \log \pi_\theta(\mathbf{o}'|\mathbf{q}') \rangle \tag{15}$$

$$= \frac{1}{TT'} \sum_{t=1}^{T} \sum_{\tau=1}^{T'} (\pi_\theta^{(t)}(\cdot|\chi) - \mathbf{e}(o_t))^\intercal \langle \nabla_\theta \mathbf{z}_t(\cdot|\chi), \nabla_\theta \mathbf{z}_\tau(\cdot|\chi') \rangle (\pi_\theta^{(\tau)}(\cdot|\chi') - \mathbf{e}(o'_\tau)) \tag{16}$$

where the middle term $\langle \nabla_\theta \mathbf{z}_t(\cdot|\chi), \nabla_\theta \mathbf{z}_\tau(\cdot|\chi') \rangle$ is empirical NTK of the language model [54].

## A.2 Extend Proposition 3.1 to Other RL Algorithms

Following previous notation, i.e., given a query with $n$ correct outputs and $N - n$ wrong outputs, the accuracy $\alpha = n/N$ and the advantages of correct and wrong output are $A_+, A_-$ respectively. We consider three RL variants:

**Dr.GRPO** [97]. The advantages are $A_+ = 1 - \frac{n}{N}, A_- = -\frac{n}{N}$. Plug-in them to Eq.(10), then

$$\nabla_\theta \mathcal{J} = \mathbb{E}_{\{\mathbf{o}_i\} \sim \pi_\theta} \frac{n(N-n)}{N^2} \left[ \frac{1}{n} \sum_{i=1}^{n} \nabla_\theta \log \pi_\theta(\mathbf{o}_{i+}|\mathbf{q}) - \frac{1}{N-n} \sum_{j=1}^{N-n} \nabla_\theta \log \pi_\theta(\mathbf{o}_{j-}|\mathbf{q}) \right] \tag{17}$$

$$= \mathbb{E}_{\{\mathbf{o}_i\} \sim \pi_\theta} \alpha(1-\alpha) \left[ \frac{1}{n} \sum_{i=1}^{n} \nabla_\theta \log \pi_\theta(\mathbf{o}_{i+}|\mathbf{q}) - \frac{1}{N-n} \sum_{j=1}^{N-n} \nabla_\theta \log \pi_\theta(\mathbf{o}_{j-}|\mathbf{q}) \right] \tag{18}$$

Then its per-step influence is to replace the original coefficient $\sqrt{\alpha(1-\alpha)}$ by $\alpha(1-\alpha)$ and other parts remain the same as GRPO.

**GPG** [98]. It multiples the advantage by a coefficient $C$ (it is $\alpha$ in original paper and we use $C$ instead to avoid ambiguity). The advantages are $A_+ = C \cdot (1 - \frac{n}{N}), A_- = C \cdot (-\frac{n}{N})$. Similar to Dr.GRPO, the coefficient in per-step influence is $C\alpha(1-\alpha)$ and other parts remain the same.

**DAPO** [99]. The advantage computation in DAPO is the same as the GRPO so the coefficient is still $\sqrt{\alpha(1-\alpha)}$. DAPO introduces other modifications such as query filtering. Therefore, the corresponding per-step influence should replace the original query set $Q$ by a filtered query set $Q'$ which only includes queries with rollout accuracy $\in (0, 1)$.

# B   More Details of Method and Experiment

## B.1   Evaluation Benchmark

We provide a query along with a vanilla demonstration answer for iGSM and PromptBench tasks. Meanwhile, we provide the corresponding DAG representation for each query for easy understanding.

**iGSM.** In iGSM, we construct a DAG by first generating the nodes, which are with the form of "each X's Y". There are two types of nodes – abstract node and instance node – based on the type of "Y". If "Y" is a class name, the node will be classified as an abstract node; if "Y" is an instance of the class, the node will be classified as an instance node. For example, consider a class "Classroom" with its two instances "Painting room", "Computer room", the "each Oakwood Middle School's Classroom" is an abstract node while "each Oakwood Middle School's Computer room" is an instance node. After generating the nodes, we then generate the dependency of instance nodes randomly as edges to construct a DAG (we do not need to add dependency to abstract nodes because it has implicit dependency, e.g., "each Oakwood Middle School's Classroom" = "each Oakwood Middle School's Painting room" + "each Oakwood Middle School's Computer room"). We can then generate query and answer based on the DAG. In query, we only list the explicit dependencies on instance nodes which may include the conditions for redundant nodes; in answer, the LLM should be able to unlock all instance and abstract nodes to the final target node with a topological order. More implementation details (e.g., how to generate random nodes and random edges, how to set the name of nodes) are systematically introduced in the original paper [74].

To improve training stability, we modified the original iGSM. Here are the main modifications: (1) we remove the modulo operation in all computation steps; (2) we filter out the queries whose ground-truth answers are larger than 1000 or smaller than −1000 to reduce the reliance on complex computation; (3) the original answer template is "Define [node name] as [a random letter], then [the random letter] = [... computation equations]". We rewrite it to increase the answer diversity while keeping the logic of problem-solving (i.e., the order of node selection) and computation details. See below for an example of the new answer template.

One example with operation number op = 10 is shown in the following text boxes. The DAG representation of this question is visualized in Figure 7.

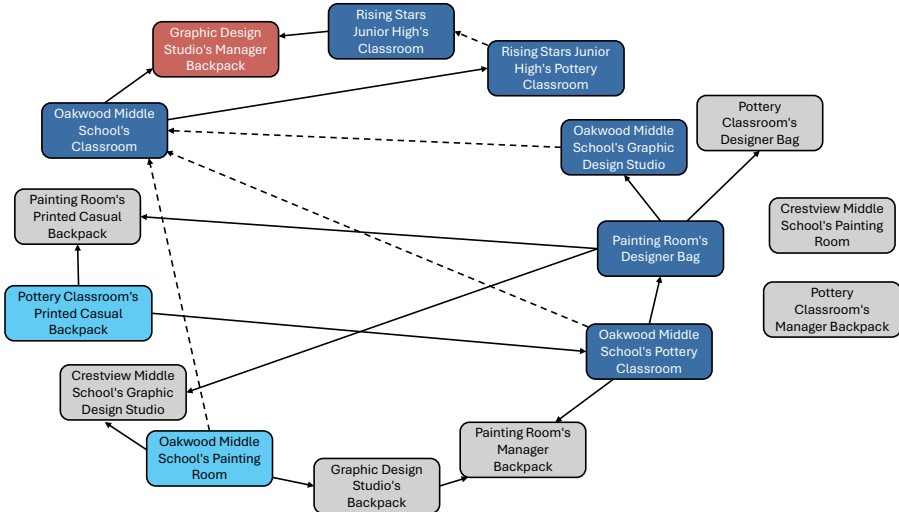

Figure 7: DAG representation of an iGSM task (number of operations = 10). The red, light blue, dark blue, and gray rectangles mean the target, leaf, intermediate, and redundant nodes, respectively. The solid arrow indicates the explicit dependency which is explicitly given in query while the dash arrow means the implicit dependency which needs to be inferred by LLM. For example, it is not directly given how to compute the value of each "Rising Stars Junior High's Classroom" in the query. The LLM is required to understand its semantic meaning and then infer that it equals to "Rising Stars Junior High's Pottery Classroom".

Note that the RL training data (15 ∼ 20 operations) and OOD test set (25 operations) in practical experiments are much harder than the example above.

**Query**:
The number of each Graphic Design Studio's Manager Backpack equals the difference of each Rising Stars Junior High's Classroom and each Oakwood Middle School's Classroom.
The number of each Painting Room's Designer Bag equals 0 more than each Oakwood Middle School's Pottery Classroom.
The number of each Painting Room's Printed Casual Backpack equals 9 more than the sum of each Painting Room's Designer Bag and each Pottery Classroom's Printed Casual Backpack.
The number of each Crestview Middle School's Graphic Design Studio equals the difference of each Painting Room's Designer Bag and each Oakwood Middle School's Painting Room.
The number of each Pottery Classroom's Designer Bag equals each Painting Room's Designer Bag.
The number of each Pottery Classroom's Manager Backpack equals 0.
The number of each Pottery Classroom's Printed Casual Backpack equals 1.
The number of each Rising Stars Junior High's Pottery Classroom equals 2 times each Oakwood Middle School's Classroom.
The number of each Oakwood Middle School's Painting Room equals 6.
The number of each Crestview Middle School's Painting Room equals 5.
The number of each Painting Room's Manager Backpack equals 0 more than the sum of each Graphic Design Studio's Backpack, each Oakwood Middle School's Pottery Classroom and each Oakwood Middle School's Painting Room.
The number of each Oakwood Middle School's Pottery Classroom equals 5 times each Pottery Classroom's Printed Casual Backpack.
The number of each Oakwood Middle School's Graphic Design Studio equals 8 more than each Painting Room's Designer Bag.
How many Manager Backpack does each Graphic Design Studio have?

---

**Vanilla CoT**:
Let's compute the answer step by step.
- According to the information given, the number of each Pottery Classroom's Printed Casual Backpack is 1. Let's denote it as S. So S = 1.
- Next, let Q represent the number of each Oakwood Middle School's Pottery Classroom. Then Q = 5 * S = 5 * 1 = 5.
- Now, we can find the number of each Painting Room's Designer Bag. Let's denote it as U. Then U = 0 + Q = 0 + 5 = 5.
- We can then calculate the number of each Oakwood Middle School's Graphic Design Studio. Let it be m. Then m = 8 + U = 8 + 5 = 13.
- The number of each Oakwood Middle School's Painting Room is 6. Let's denote it as W. So W = 6.
- Then, let's denote the number of each Oakwood Middle School's Classroom as v. Then v = m + Q + W = 13 + 5 + 6 = 24.
- We can then calculate the number of each Rising Stars Junior High's Pottery Classroom. Let it be B. Then B = 2 * v = 2 * 24 = 48.
- Now, we can find the number of each Rising Stars Junior High's Classroom. Let it be p. Then p = B = 48.
- Next, let y represent the number of each Graphic Design Studio's Manager Backpack. Then y = p - v = 48 - 24 = 24.
Thus, the answer is 24.

---

**PromptBench.** PromptBench generates data in two stages: (1) DAG construction and (2) Natural language description of the DAG. For DAG construction, we first generate a directed acyclic graph (DAG) with a specified depth and number of redundancies. The DAG is constructed top-down: we begin by generating the root node and then recursively sampling dependencies between each node and its parent node(s). If a node uses a binary operator, it has two parent nodes; if it uses a unary operator, it has one parent node. This process continues recursively until the desired depth is reached. During DAG construction, each node is assigned a unique name generated by a random string generator. Once the DAG structure is complete, we sample values for all leaf nodes from a predefined set and compute the values of internal nodes in a bottom-up manner. For the natural language description stage, we describe the constructed DAG and its associated computation using predefined templates to generate a textual problem description. To improve the training stability, we also filter out the queries whose ground-truth answers are larger than $1000$ or smaller than $-1000$. Full details of the generation process can be found in the original PromptBench paper [86].

One example with depth = 4, number of redundancy = 2 is shown in the following text boxes. The DAG representation of this question is visualized in Figure 8.

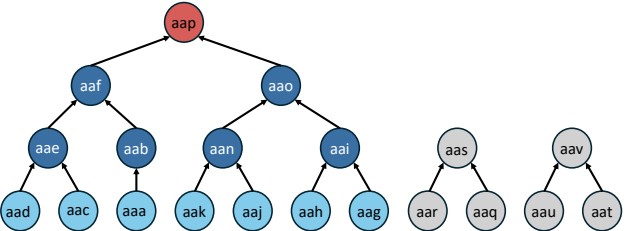

Figure 8: DAG representation of a PromptBench task. The red, light blue, dark blue, and gray circles mean the target, leaf, intermediate, and redundant nodes, respectively.

---

**Query**:
The value of aac is 5.
aaf gets its value by adding together the value of aab and aae.
The value of aaj is 5.
The value of aak is 9.
The value of aar is 2.
The value of aaa is 9.
The value of aat is 3.
aao gets its value by adding together the value of aai and aan.
The value of aad is 5.
aas gets its value by multiplying together the value of aaq and aar.
The value of aaq is 5.
aan gets its value by adding together the value of aak and aaj.
The value of aau is 10.
aai gets its value by multiplying together the value of aah and aag.
aap gets its value by subtracting the value of aao from the value of aaf.
The value of aah is 2.
aav gets its value by multiplying together the value of aat and aau.
aab gets its value by squaring the value that aaa has.
The value of aag is 6.
aae gets its value by subtracting the value of aac from the value of aad.
What is the value of aap?

---

**Vanilla CoT**: Let's compute the answer step by step.
Let's solve aaa, aaa is 9
Let's solve aab, aab = aaa$^2$ = 81
Let's solve aac, aac is 5
Let's solve aad, aad is 5
Let's solve aae, aae = aac - aad = 0
Let's solve aaf, aaf = aab + aae = 81
Let's solve aah, aah is 2
Let's solve aaj, aaj is 5
Let's solve aag, aag is 6
Let's solve aai, aai = aag * aah = 12
Let's solve aak, aak is 9
Let's solve aan, aan = aaj + aak = 14
Let's solve aao, aao = aai + aan = 26
Let's solve aap, aap = aao - aaf = 55
Thus, the answer is 55.

---

**Why are iGSM and Promptbench good testbeds for LLM reasoning evaluation?** We can observe that both tasks require the problem-solving capabilities of LLM from multiple perspectives, e.g., basic arithmetic calculation, search and planning, which mirrors reasoning tasks in realistic scenarios. Meanwhile, these tasks require few priors (it only needs very basic concept understanding in iGSM task) and focus on the **abilities** instead of **knowledge**. Moreover, all the data are synthetic and the queries are almost infinite (see estimation in [74]). Therefore, it avoiding the data contamination,

which can significantly confound the experiment results, while keeping substantial diversity of the query set.

## B.2 The Extraction of DAG representation

For the datasets we used in this paper (iGSM and promptbench), all questions and answers are rule-based and follow a fixed sentence structure, which allows us to extract the DAG representation through simple **string matching**.

Take an iGSM problem for example, each variable mentioned in the question is treated as a node, and we define an edge from one node to another if the value of the former depends on the latter. Consider a premise *The number of each Painting Room's Printed Casual Backpack equals 9 more than the sum of each Painting Room's Designer Bag and each Pottery Classroom's Printed Casual Backpack*, we view *each Painting Room's Printed Casual Backpack* as a node. Since it depends on nodes *each Painting Room's Designer Bag* and *each Pottery Classroom's Printed Casual Backpack* and we draw edges from these two nodes to the former. By iterating all the premises in this way, we construct the node set V and edge set E of the graph. Similarly, we can also apply the same procedure to the answer. However, we will miss the redundant nodes if the answer only includes the minimal topological path to the final node.

For other QA datasets without a fixed format, we can employ an oracle LLM (e.g., GPT4) to parse the node and edge from the QA. Specifically, we can view all intermediate variables / conclusions / corollaries as node and the edge is still their dependency. Here is a prompt to extract $(V, E)$:

---

**Prompt to extract DAG**:
You are a helpful data analyst. You will be given a question-answer pair. Your task is to extract the DAG of the question. Here are some instructions for extracting the DAG:
- First separate the answer into multiple steps. If one step includes multiple intermediate variables / conclusions, further separate it until each step includes only one.
- Identify NODE in each step. The node can be intermediate variables / conclusions / corollary.
- Identify the dependency between the nodes.
- Remember that the final graph should be acyclic.
You should return the DAG in the following format:
- the name of the nodes, e.g., node 1: x, node 2: y, node 3: z
- the dependent list of each node, e.g., node 1: [], node 2: [1], node 3: [1, 2]
[few-shot examples]
[QA]

---

## B.3 Examples of Injected Behaviors

In this section, we give an example of injected behaviors on iGSM task.

---

**Injected behaviors in iGSM**:
Let's compute the answer step by step.
...
- The number of each Oakwood Middle School's Painting Room is 6. Let's denote it as W. So W = 6.
- Then, let's denote the number of each Oakwood Middle School's Classroom as v. Then v = m + Q + W = 13 + 5 + 6 = 18 + 6 = 24.
...
- Then, let's denote the number of each Graphic Design Studio's Manager Backpack as y. But we haven't calculated the number of each Rising Stars Junior High's Classroom yet, thus the value of y is still unknown.
...
- Now, we can find the number of each Graphic Design Studio's Manager Backpack. Remember that it has been denoted as y. We know that it equals the difference between each Rising Stars Junior High's Classroom and each Oakwood Middle School's Classroom. Then y = p - v = 48 - 24 = 24.
Thus, the answer is 24.

---

The blue, violet, and cyan parts correspond to the subgoal computation, reflection, and information analysis behaviors. In subgoal computation, we enforce LLM to conduct only one operation (i.e., $+, -$ or $\times$) in every derivation and keep all intermediate computations for complex equations. In reflection, the LLM explores a locked (not solvable yet) node "each Graphic Design Studio's Manager Backpack" and then goes back. In information analysis, the LLM integrates the information gathered from the query for better derivation of computation, which is also an exploitative behavior.

### B.4 Training Details

**SFT dataset and baseline implementations**. For the iGSM task, we use 2000 SFT data for `Vanilla` and BRIDGE. The `PP-Aug` and `RC-Aug` augment each data for additional three times so they have 8000 SFT data in total. For the PromptBench task, we use 5000 SFT data for `Vanilla` and BRIDGE. The `PP-Aug` and `RC-Aug` augment each data for an additional once so they have 10000 SFT data in total.

**SFT training**. We first train the LLMs by SFT to narrow the domain gap between the pretraining corpus and evaluation tasks and enforce the model to answer the question with the demonstrated template, where the final reward is easy to extract and verify. The other training configurations are attached below, which are shared by all base models on both the iGSM and PromptBench tasks.

Table 3: Configurations of SFT training

| Configurations | value |
|---|---|
| training epoch | 5 |
| batch size | 128 |
| learning rate | $5 \times 10^{-6}$ |
| learning rate scheduler | constant |

The system prompt along with query and answer templates are shown as follows:

---
**System prompt**:
A conversation that the assistant solves the user's problem. The assistant first thinks about the reasoning process in the mind and then provides the user with the answer. The reasoning process and answer are enclosed within <think> </think> and <answer> </answer> tags, respectively, i.e., <think> all the reasoning process here </think> <answer> final answer here </answer>.

---
**SFT data template** (take Qwen tokenizer for example)
<lim_start|>system<lim_end|>
{system prompt}
<lim_start|>user<lim_end|>
{query}
<lim_start|>assistant<lim_end|>
<think> {CoT answer} </think>

<answer> The final answer is | {final answer} | </answer>

---

**RL training**. After the SFT stage, we apply RL to further fine-tune the SFT models. Our implementation is based on VeRL [100]. When computing KL divergence, we use the low variance implementation [101], aligning with the GRPO implementation [23]. The shared parameters are listed in Table 4.

In rollout, we set the maximum generation length as 2560 for iGSM and 1536 for PromptBench. We use top $p = 1.0$ for sampling in generation. Besides, we observe that Llama3.2-1B may deviate from answer template so we add a format reward to it, i.e., it will obtain a 0.05 reward if it strictly follows the given answer template (i.e., "<think> ... </think> <answer> ... </answer>") addition to the correctness reward.

### B.5 More Experiment Results

We attach the training curves of main experiments in Fig. 9&10.

Table 4: Configurations of RL training

| Configurations | value |
| --- | --- |
| batch size | 256 |
| learning rate | $1 \times 10^{-6}$ |
| learning rate scheduler | constant |
| rollout number per query $N$ | 32 |
| rollout temperature | 1.0 |
| rollout backend | vllm [102] |
| KL coefficient $\beta$ | 0.001 |

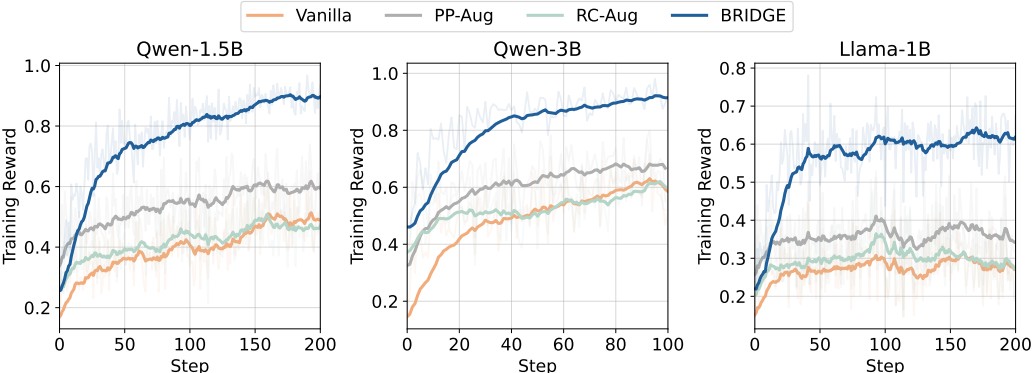

Figure 9: The training curve of experiments in the iGSM task.

## B.6 Comparison between BRIDGE and Accuracy-based Rejection Sampling

Based on the analysis of two factors affecting the performance in RL in Proposition 3.1, there is an intuitive idea that filters out queries with excessively high or low rollout accuracy (close to 1 or 0) as discussed in the beginning of sec. 3.3. In this section, we compare the performance of our method with the rejection-sampling-based RL. Specifically, we roll out the SFT models (pre-RL models) for 8 times on a larger dataset that shares the same distribution as the original RL training set. We then retain only the queries with 1 to 7 correct answers, resulting in a filtered RL training dataset with a medium accuracy ratio. Note that the new dataset has the same size as previous RL training dataset. Then we run RL on the rejection sampled dataset starting from the same SFT model. The results are shown in Fig. 11.

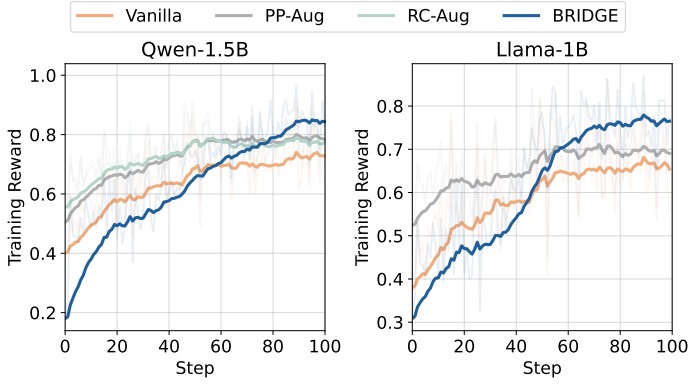

Figure 10: The training curve of experiments in the PromptBench task.

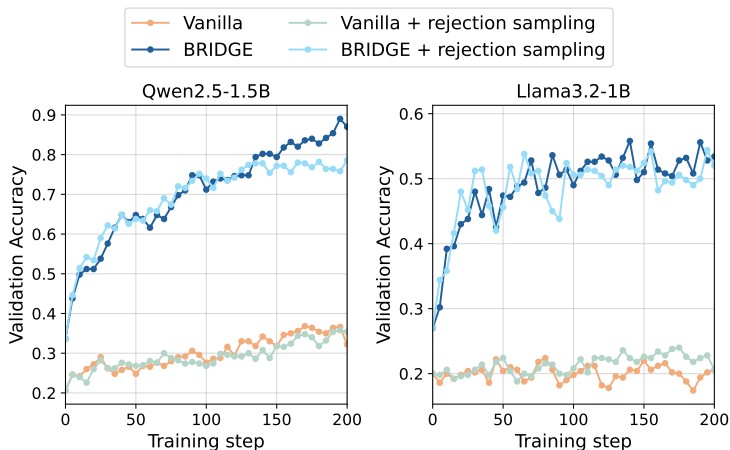

Figure 11: The comparison of BRIDGE with rejection-sampling-based filtering.

We can observe that the rejection sampling does not improve the RL performance over `Vanilla` baseline and is significantly worse than BRIDGE. Meanwhile, when applying rejection sampling to the BRIDGE, the difference is very minor, and it even degrades the final performance on the Qwen-1.5B model. The reason for its poor performance is that rejection sampling can only shape the accuracy distribution of RL rollouts at the initial stage. As training progresses, the distribution may shift and is not guaranteed to remain within the medium-accuracy range. More importantly, this method does not improve the data co-influence. Based on the above results, we can conclude that it is more effective to improve the data co-influence when preparing LLM for RL, which relies on behavior injection, rather than simply adjusting rollout accuracy by distorting the query distribution.

## B.7 Details of Data Co-influence and Per-step Influence Estimation

Here is a more detailed description on the computation of per-step influence shown in Fig. 3 right.

1. Begin by 2,000 queries. For each query, we generate 8 rollouts using the model, resulting in a total of 16,000 query–output pairs. Each query is then assigned to an accuracy group based on the accuracy of its rollouts.

2. For each query-output pair, we compute 1) advantage, and 2) the policy gradient $\nabla_\theta \log \pi_\theta(o|q)$.

3. Using Eq.(5), we estimate the per-step influence for each query $q$. In this formulation, $(q', o')$ ranges over all 16,000 query–output pairs.

4. We then group queries by their accuracy (0/8, 1/8, ..., 8/8), and compute the average per-step influence for each group. These results are visualized in Fig. 3 right. Notably, the groups with 0/8 and 8/8 accuracy have zero per-step influence because the corresponding advantages are zero under the GRPO formulation.

Since computing the inner product $\mathcal{K}_\theta[.,.]$ is computationally intractable, we mainly follow LESS [82] in data co-influence estimation. Specifically, there are two main steps reducing the dimensionality of $\nabla_\theta \log \pi_\theta(\mathbf{o}|\mathbf{q})$: (1) we first compute the LoRA gradient $\nabla_{\theta_{\text{LoRA}}} \log \pi_{\theta_{\text{LoRA}}}(\mathbf{o}|\mathbf{q})$ rather than the full-parameter gradient by backpropagating the likelihood loss to the LoRA modules. (2) Then we apply random projection on the LoRA gradient and get a vector with a smaller dimension, which preserves the inner product of two gradients [103]. For the LoRA module, we specified a rank of 64, an $\alpha$ value of 128, a dropout rate of 0.1, and learned LoRA matrices for all attention matrices. For the random projection, the final dimension of the projection output is $8192$. Then we can estimate the data co-influence coefficient $\mathcal{K}_\theta[\cdot, \cdot]$ by the inner product between two query-output samples.

After obtaining the data co-influence coefficient, we can compute the per-step influence $\Delta \mathcal{J}$ by plugging the data co-influence and advantages. **Note that we did not multiple the learning rate $\eta$ when computing the results in Fig. 3.**

**B.8 Computation Overhead**

All experiments can be run on a server with $2\times$A100 (80G). Each SFT experiment takes less than 1h. Regarding the RL experiments, it takes $\sim$ 8h for Qwen-1.5B and Llama-1B to complete 200-step RL and $\sim$ 6h for Qwen-3B to run 100-step RL on iGSM while it spends $\sim$ 2h to run RL on PromptBench (100 steps) for Qwen-1.5B and Llama-1B due to shorter rollout length.

