# OpenReview forum: "Behavior Injection: Preparing Language Models for Reinforcement Learning"
_NeurIPS.cc/2025/Conference — NeurIPS 2025 poster_

### Official Review · Reviewer_SDEM · 2025-06-02

**Clarity:** 2
**Significance:** 2
**Originality:** 3
**Rating:** 4
**Confidence:** 4

**Summary:**

This paper introduces, BRIDGE, an algorithm to add data to the SFT phase before performing RL training. Specifically, it converts question answer pairs to directed acyclic graph. Using the graph structure, the authors identify locked and unlocked nodes which are used to craft exploration and exploitation behaviors. They compare the BRIDGE algorithms to several baselines for multiple base models on 2 tasks. They use the notion of per-step influence function to analyze their algorithm and the baselines.

**Questions:**

* How do you extract DAG G = (V, E) from (q, a)?
* How do you identify locked and unlocked nodes?
* Could BRIDGE be applied throughout RL training and not only prior?

**Ethical Concerns:**

["NO or VERY MINOR ethics concerns only"]

**Final Justification:**

The authors addressed my main concern that their method would not scale to non synthetic data. The demonstrated that their method could be applied to GSM8k and show great improvement. I would encourage the author to rewrite the paper to focus more on GSM8k than on the synthetic datasets.

**Limitations:**

From my current understanding of the method, the limitations are not well described. For example, could BRIDGE be used on other math and common sense reasoning benchmarks such as GSM8k and MATH?

**Quality:**

2

**Strengths And Weaknesses:**

## Strengths

* Using per-step influence function to analyse RL training is quite interesting. I wish the authors analyzed this quantity in more details and throughout training.
* The results on both iGSM and PromptBench are strong for multiple base models.

## Weaknesses

### Clarity

* > 1: Extract DAG G = (V, E) from (q, a).

It is very unclear how to transform any question-answer pair into a DAG.

* > We generate exploration behaviors by attempting to solve a locked (i.e., not yet solvable) node ahead of time, followed by reflection. For exploitation behaviors, we aggregate available information to solve an unlocked node (i.e.,solvable but not yet solved), reinforcing the known reasoning path

It is unclear how to identify locked and unlocked nodes

* Figure 2: no legend for darker nodes.

* Figure 3: why are the width of the bar not all the same?

* The connection between per-step influence and BRIDGE is unclear.

* Figure 3 Right should be discussed in more details.

### Benchmarking

iGSM and prompt bench are both synthetic and uncommon benchmarks. Could the paper be extended to more common benchmarks such as GSM8k?

---

> ### Author Rebuttal · Authors · 2025-07-31
>
> We thank the reviewer for your review and feedback. Here are our responses:
> - **W1 & Q1. How do you extract DAG from QA**
>
>   Thanks for pointing it out. We provide more details about the extraction process here.
>
>   For the datasets we used in this paper (iGSM and promptbench), all questions and answers are rule-based and follow a fixed sentence structure, which allows us to extract the DAG representation through simple **string matching**. In iGSM problem, each variable mentioned is treated as a node and the edge is the dependency between nodes. Consider a premise `The number of each Painting Room’s Printed Casual Backpack equals 9 more than the sum of each Painting Room’s Designer Bag and each Pottery Classroom’s Printed Casual Backpack.`. we view `each Painting Room’s Printed Casual Backpack` as a node. Since it depends on nodes `each Painting Room’s Designer Bag` and `each Pottery Classroom’s Printed Casual Backpack` and we draw edges from these two nodes to the former. By iterating all premises in this way, we construct the node set V and edge set E of the graph. We also provide a complete example in Appendix B.1.
>
>   For other QA datasets without a fixed format, we can employ an oracle LLM (e.g., GPT4) to parse the node and edge from the QA. Specifically, we can view all intermediate variables / conclusions / corollaries as node and the edge is still their dependency. Here is an example prompt to extract (V, E):
>   ```
>   You are a helpful data analyst. You will be given a question-answer pair. Your task is to extract the DAG of the question. Here are some instructions for extracting the DAG:
>   - First separate the answer into multiple steps. If one step includes multiple intermediate variables / conclusions, further separate it until each step includes only one.
>   - Identify NODE in each step. The node can be intermediate variables / conclusions / corollary.
>   - Identify the dependency between the nodes.
>   - Remember that the final graph should be acyclic.
>
>   [few-shot examples]
>   [QA]
>   [format instruction]
>   ```
>   Given a QA sample from MATH dataset:
>   ```
>   Question:
>   Define \[ p = \sum_{k = 1}^\infty \frac{1}{k^2} \quad \text{and} \quad q = \sum_{k = 1}^\infty \frac{1}{k^3}. \]
>   Find a way to write \[ \sum_{j = 1}^\infty \sum_{k = 1}^\infty \frac{1}{(j + k)^3} \] in terms of $p$ and $q$.
>
>   Answer:
>   We count the number of times $\frac{1}{n^3}$ appears in the sum \[ \sum_{j = 1}^\infty \sum_{k = 1}^\infty \frac{1}{(j + k)^3}\], where $n$ is a fixed positive integer. (In other words, we are conditioning the sum on $j + k$).
>   We get a term of $\frac{1}{n^3}$ each time $j + k = n.$ The pairs $(j,k)$ that work are $(1,n - 1),(2,n - 2),\dots,(n - 1,1)$ for a total of $n - 1$ pairs.
>   Therefore,
>   \begin{align*}
>   \sum_{j = 1}^\infty \sum_{k = 1}^\infty \frac{1}{(j + k)^3} &= \sum_{n = 1}^\infty \frac{n - 1}{n^3} \\
>   &= \sum_{n = 1}^\infty \left( \frac{n}{n^3} - \frac{1}{n^3} \right) \\
>   &= \sum_{n = 1}^\infty \left( \frac{1}{n^2} - \frac{1}{n^3} \right) \\
>   &= \sum_{n = 1}^\infty \frac{1}{n^2} - \sum_{n = 1}^\infty \frac{1}{n^3} \\
>   &= \boxed{p - q}.
>   \end{align*}
>   ```
>   The extract result is:
>   ```
>   Node names:
>   node 1: ( S = \sum_{j = 1}^\infty \sum_{k = 1}^\infty \frac{1}{(j + k)^3} )
>   node 2: Number of pairs ((j, k)) with (j + k = n) is ( n-1 )
>   node 3: ( S = \sum_{n=1}^\infty \frac{n-1}{n^3} )
>   node 4: ( \frac{n-1}{n^3} = \frac{1}{n^2} - \frac{1}{n^3} )
>   node 5: ( S = \sum_{n=1}^\infty \left(\frac{1}{n^2} - \frac{1}{n^3}\right) )
>   node 6: ( S = \sum_{n=1}^\infty \frac{1}{n^2} - \sum_{n=1}^\infty \frac{1}{n^3} )
>   node 7: ( S = p - q )
>
>   Dependency list:
>   node 1: []
>   node 2: []
>   node 3: [1, 2]
>   node 4: []
>   node 5: [3, 4]
>   node 6: [5]
>   node 7: [6]
>   ```
>   We will update the details and make it more clear in revision.
>
> - **W2 & Q2. How do you identify locked and unlocked nodes?**
>
>   With the extracted DAG, we construct the solution step by step while keeping track of the solved nodes. A locked node is one whose parent nodes have not all been solved yet. In contrast, an unlocked node is a node that has not yet been solved, but all of its parent nodes are already resolved.
>
> - **W3. Figure 2: no legend for darker nodes.**
>
>   The blue node generally indicates any non-target node. The dark blue node indicates the intermediate node that has parent node, while the light blue indicates the source node that does not have parent node.
>
> - **W4. Figure 3: why are the width of the bar not all the same?**
>
>   The bars differ in width because the middle bar represents all rollouts with accuracy in the range $\in (0, 1)$, which covers multiple discrete accuracy $1/8, ..., 7/8$ and is therefore wider than the left bar (accuracy = 0) and right bar (accuracy = 1). We highlight the $(0, 1)$ interval separately because these rollouts have non-zero advantages and per-step influence. This visualization emphasizes the distribution and impact of partially correct rollouts on the model’s improvement.
>
> - **W5 & W6. The connection between per-step influence and BRIDGE is unclear. Figure 3 Right should be discussed in more details.**
>
>   Thanks for bringing up the concerns. Let's begin with fig.3 right. The computation of per-step influence is briefly described in Lines 250–259. We provide additional details:
>   - Begin by 2,000 queries. For each query, we generate 8 rollouts using the model, resulting in a total of 16,000 query–output pairs. Each query is then assigned to an accuracy group based on the accuracy of its rollouts.
>   - For each query-output pair, we compute 1) advantage, and 2) the policy gradient $\nabla_\theta \log \pi_\theta (o|q)$.
>   - Using Eq.(5), we estimate the per-step influence for each query $q$. In this formulation, $(q', o')$ ranges over all 16,000 query–output pairs.
>   - We then group queries by their accuracy (0/8, 1/8, ..., 8/8), and compute the average per-step influence for each group. These results are visualized in fig.(3) right. Notably, the groups with 0/8 and 8/8 accuracy have zero per-step influence because the corresponding advantages are zero under the GRPO formulation.
>   - Since computing the inner product $\mathcal{K}_\theta[.,.]$ is computationally intractable, we apply dimension reduction following previous work. More details are in Appendix B.6.
>
>   The results demonstrate that BRIDGE increases the per-step influence, contributing to more effective RL updates and explains the improved performance we observe after fine-tuning.
>
>   For the first concern "The connection between per-step influence and BRIDGE is unclear", we respectfully disagree with it. As we derive in line 107~109, the per-step influence measures how model performance changes after training on a specific query with its outputs. We evaluate the per-step influence of different SFT models and present the experiment results in fig.3 right. The results clearly show that BRIDGE consistently exhibits higher per-step influence. This directly supports our claim in **sec.3.2 takeaway** that BRIDGE provides more effective learning signals during RL training, resulting in larger performance gains compared to other baselines.
>
> - **W7. iGSM and prompt bench are both synthetic and uncommon benchmarks. Could the paper be extended to more common benchmarks such as GSM8k?**
>
>   Yes. To extend to GSM8K, we apply our DAG extraction method to 2,000 examples from the GSM8K training set. Using the extracted DAGs, we leverage an oracle LLM (GPT4.1) to augment the data by injecting information analysis and reflection behaviors into the answer CoT. We then do SFT on Qwen2.5-1.5B and Llama3.2-1B models for 5 epochs and use the remaining training data for RL (150 steps). The results are listed below:
>   | Base model|  | Vanilla | BRIDGE   |
>   |---| --- | --- | -------- |
>   | Qwen 1.5B | SFT  | 55.8    | 61.6 |
>   |   | RFT      | 63.5 | **77.1** |
>   |   | $\Delta$ | 7.7  | **15.5** |
>   | Llama 1B  | SFT | 44.2    | 46.8 |
>   |   | RFT      | 48.6 | **59.6** |
>   |   | $\Delta$ | 4.4  | **12.8** |
>
>   Moreover, we want to emphasize that while evaluating on real-world benchmarks like GSM8K is important, synthetic benchmarks offer unique advantages as testbeds for RL research: 1) Synthetic benchmarks focus on to structured reasoning and problem-solving ability, minimizing the confounding effects of background knowledge, which is required by many public benchmarks but ususally lacked by some LLMs. 2) There is a severe data contamination of public benchmarks in many LLMs, as suggested by recent works [1][2], while the synthetic data fundamentally prevent it.
>
> - **Q3. Could BRIDGE be applied throughout RL training and not only prior?**
>
>   Thank you for this suggestion. It could be a promising direction for future research. While our current work uses BRIDGE prior to RL, we agree it could be applied iteratively during training. For example, one could periodically pause RL, use the partially trained model to generate improved reasoning traces, and then apply BRIDGE to synthesize new training examples for an intermediate round of supervised fine-tuning before resuming RL (which is also mentioned in Deepseek R1 paper [3]). We believe such iterative training can be another learning paradigm to further boost performance and improve sample efficiency. We will add a discussion of this potential extension in the revised manuscript.
>
> We hope our detailed responses and new experimental results have fully addressed the reviewer's concerns. We would be grateful for their reconsideration of our manuscript in light of these clarifications.
>
> [1] Shao, Rulin, et al. "Spurious rewards: Rethinking training signals in rlvr."
>
> [2] Wu, Mingqi, et al. "Reasoning or Memorization? Unreliable Results of Reinforcement Learning Due to Data Contamination."
>
> [3] Guo, Daya, et al. "Deepseek-r1: Incentivizing reasoning capability in llms via reinforcement learning."

---

### Official Review · Reviewer_to1g · 2025-06-29

**Clarity:** 3
**Significance:** 4
**Originality:** 3
**Rating:** 5
**Confidence:** 4

**Summary:**

This paper investigates why reinforcement fine-tuning (RFT) sometimes yields large gains for large language models (LLMs) and sometimes fails to help. The authors identify two key conditions that make a model RL-ready: (1) an intermediate rollout accuracy (the base model’s initial success rate on task rollouts should be neither too low nor too high) and (2) a strong data co-influence (training on one sample should generalize to others, indicating the model can propagate learned knowledge). To prepare an LLM to better benefit from RFT, the paper proposes Behavior Injection, a data-centric augmentation strategy called BRIDGE. The idea is to enrich the supervised fine-tuning (SFT) data with behaviors that encourage exploration and exploitation before any RL training. Concretely, the authors augment the model’s chain-of-thought demonstrations with additional reasoning steps: subgoal computation (explicitly breaking down problems into intermediate steps), information analysis (aggregating relevant info in reasoning), and reflection (exploring an alternative path or revisiting a sub-problem). By seeding these behaviors in the SFT phase, the base model produces more informative and varied trajectories during RL, which in turn leads to larger performance improvements from the RL fine-tuning. The paper evaluates BRIDGE on two challenging reasoning benchmarks (grade-school math iGSM and the arithmetic reasoning subset of PromptBench), using multiple base models of different sizes. In experiments, models prepared with behavior injection consistently achieve significantly higher gains from RFT compared to baseline approaches (including no augmentation, premise shuffling, and existing chain-of-thought augmentations). For example, BRIDGE-prepared models not only start with solid performance but after RL fine-tuning they outperform others by a large margin on both in-distribution and out-of-distribution test sets. Overall, the paper’s contributions include a theoretical analysis of RL dynamics for LLMs, a novel data augmentation method to improve those dynamics, and empirical evidence that this method makes LLMs more amenable to reinforcement learning.

**Questions:**

1. How well does behavior injection generalize to more complex or interactive tool-use tasks beyond static QA or math problems?

2. What’s the impact of the demonstration source or quality? Were they human-written, model-generated, or noisy? How robust is the method to imperfect traces?

3. How does behavior injection compare to prompting approaches like “think step-by-step” at inference time? Does fine-tuning offer clear advantages?

4. Are there cases where the model overuses tools unnecessarily? Any error analysis on when behavior injection might lead to misuse would be helpful.

**Ethical Concerns:**

["NO or VERY MINOR ethics concerns only"]

**Limitations:**

The authors explicitly discuss some limitations and potential negative impacts, which is commendable. They acknowledge that the scope of evaluation is limited to math and common-sense reasoning tasks, and suggest that future work should extend BRIDGE to broader domains (e.g. more complex agent environments) and with larger datasets. I agree with this assessment – the method’s current validation, while thorough for what it is, doesn’t cover all possible use cases. Another limitation implicitly noted is that the approach thus far is demonstrated on tasks with ground-truth verifiable rewards; tasks involving subjective or human-provided rewards are not directly tested, which could be addressed in the future. The authors also mention a potential negative societal impact: the misuse of their method. Since behavior injection is a data augmentation technique, a malicious actor could in principle inject undesirable behaviors into a model (for example, tendencies towards biased, unethical, or unsafe outputs) and then amplify them via RL. This is a realistic concern – if someone “prepares” a model with harmful behaviors and then optimizes for them with RL, it could yield a more capable but harmful agent. The paper advises caution here, which is appropriate. Mitigating this risk would involve careful curation of behaviors and perhaps oversight during RL (an issue that overlaps with general AI safety considerations). On the positive side, the method is aimed at improving reasoning and alignment with correct answers, which is a beneficial goal; there is nothing inherently dangerous in BRIDGE itself if used responsibly. One limitation not explicitly discussed in the paper (but worth noting) is the manual effort or expertise required to design injected behaviors. As raised above, BRIDGE currently relies on human intuition to decide what constitutes a helpful exploratory or exploitative behavior for a task. This could be seen as a form of expert data augmentation. If a user of this method does not have a good understanding of the task’s solution structure, they might struggle to apply BRIDGE effectively. While not a dire flaw, this is a practical consideration: the need for domain-specific behavior engineering could limit adoption or require further research (perhaps the authors could mention this as an area for future improvement). Aside from that, I did not identify major overlooked issues. The paper is quite conscientious in evaluating its own scope. Computational cost is reasonably addressed (the method doesn’t drastically increase training cost, aside from some overhead to generate augmented data and compute influence metrics for analysis). The ethical considerations are relatively mild for this work (since it’s about improving model training, not directly producing problematic content). The mention of misuse covers the main worry. In sum, the authors have adequately discussed limitations and societal impacts, and I appreciate their transparency. Any omissions are minor and can be tackled in subsequent research or the rebuttal.

**Paper Formatting Concerns:**

None.

**Quality:**

4

**Strengths And Weaknesses:**

Strengths:
Quality (analysis): The paper provides a thoughtful theoretical analysis of what drives learning in the RL fine-tuning stage. It introduces a per-step influence framework that isolates how rollout accuracy and a co-influence coefficient contribute to gradient signal during RFT. This analysis yields an intuitive insight: if a model gets every training query either entirely correct or entirely wrong, the RL reward signal is uninformative (gradient is zero in those extreme cases). The most learning happens when the model’s initial answers are partially correct. This perspective is well-grounded and novel in the LLM context, and it directly motivates the proposed solution (shape the SFT model such that its knowledge is incomplete but improvable under RL). The analytical findings are clearly connected to the method, lending a strong foundation to the paper’s approach.
Originality: The idea of “behavior injection” to make an LLM more RL-ready is innovative. Rather than modifying the RL algorithm or reward scheme, the authors take a data-centric approach: they augment the training data to instill certain behaviors before performing RL. While data augmentation for better reasoning is not new, this work is the first (to my knowledge) to explicitly inject exploratory and exploitative reasoning patterns to facilitate subsequent policy optimization. This behavior-level augmentation goes beyond simply adding more data or shuffling it – it changes how the model approaches problems (e.g. sometimes pausing to reflect or breaking problems into subparts). This approach of prepping a model for RL by curating its reasoning habits is a fresh contribution and could inspire other “RL-preparation” techniques.
Significance: The ability to improve an LLM’s performance via RL with less hassle is quite significant. Many important applications of LLMs (e.g. solving complex math problems, planning, tool use) rely on reinforcement fine-tuning for alignment or enhanced reasoning. However, RFT can be unstable or yield minimal gains if the base model isn’t already in a good state. This paper’s findings and method address that bottleneck. Empirically, the gains reported are impressive: for instance, on the iGSM math benchmark, the BRIDGE-augmented models see large accuracy jumps after RL (up to ~30-50% absolute improvement over their pre-RL accuracy, far exceeding baseline gains). Importantly, these gains hold on both seen (in-distribution) and harder (out-of-distribution) problems, indicating robustness. Such improvements suggest that BRIDGE could make RL fine-tuning a more routinely effective tool, potentially impacting how practitioners approach alignment and domain adaptation of LLMs. Even if the method is demonstrated on math and logic tasks here, the general idea could be applicable to a wide range of reasoning-heavy domains.
Quality (experiments): The experimental evaluation is thorough. The authors test their method on two quite different reasoning datasets (one focused on multi-step math, another on logical/arithmetic reasoning with varying irrelevant information) using two model families (Qwen and LLaMA, with multiple model sizes). They compare against strong augmentation baselines – e.g. PP-Aug (premise permutation) and RC-Aug (reasoning chain augmentation from prior work) – rather than just against a vanilla model. This shows the authors have situated their work in context and are pushing beyond existing techniques. The results consistently favor BRIDGE, and the paper reports not just final accuracy but also the performance gain due to RL (“Delta by RFT” in tables), which directly measures how much RFT helped in each case. This is a appropriate metric given the paper’s focus. Additionally, the authors include ablation studies: they try injecting each behavior (reflection, subgoaling, analysis) in isolation to gauge their individual contributions. These ablations (illustrated in their Figures 4–6) reveal that each injected behavior does help (e.g. reflection consistently boosts final performance, while analysis primarily speeds up learning) and that the combination (BRIDGE) is most effective for robust gains. Such detailed empirical analysis strengthens confidence in the approach. The paper also measures proxy metrics like model perplexity and action entropy, showing that the injected behaviors indeed increased the model’s tendency to explore (higher entropy with reflection) and exploit (lower perplexity with step-by-step reasoning). Overall, the experiments are convincing and well-designed.
Clarity: The paper is well-written and structured in a logical manner. The problem context is explained succinctly (with relevant references to prior work on RL for LLMs and data augmentation). The introduction clearly states the motivation and contributions without overstating them. Technical sections are dense but generally understandable: key new terms like “data co-influence coefficient” are defined, and there’s an effort to provide intuition (e.g. Proposition 3.1 and the discussion around it give insight into how initial accuracy and cross-sample influence affect learning). The inclusion of figures (such as the overview diagram in Figure 1 and the influence visualization in Figure 3) helps the reader grasp the concepts. The supplementary material appears to provide useful extras (e.g. examples of the augmented behaviors in Appendix B.2 and training details in B.3), which shows the authors’ care in ensuring reproducibility and clarity. Minor points: there are a few notational heavy parts (the equations for influence may be hard to parse on a first read), but the authors do summarize key takeaways in plain language (e.g. end of Section 4.3, they summarize what Figures 3 showed). In sum, the clarity is sufficient for a NeurIPS submission, and the professional tone and organization make it accessible to the target audience (researchers familiar with LLMs and RL).

Weakness:

While the paper is strong overall, a few areas could be clarified or expanded to enhance impact and applicability:

Limited Domain Scope: The experiments are limited to math and structured reasoning tasks with programmatically verifiable rewards. While well-chosen for controlled analysis, it remains unclear how well BRIDGE generalizes to domains like dialogue, code generation, or interactive environments where rewards may be noisier or more subjective. The authors acknowledge this and suggest it as future work, but even a discussion on how to adapt BRIDGE to such settings would improve confidence in its broader applicability.

Manual Behavior Design: The effectiveness of BRIDGE depends on hand-crafted behaviors (e.g., reflection, subgoaling) tailored to specific tasks. This may limit generalization unless clearer guidelines or semi-automated methods for behavior selection are provided. Including such guidance would improve the method’s transferability across domains.

Base Model Dependence: BRIDGE assumes a reasonably capable base model that can already express chain-of-thought reasoning. It’s unclear how the method would perform with significantly weaker or much larger models. Some empirical or theoretical discussion on how model capacity interacts with the benefits of behavior injection would be helpful.

Empirical Scope and Metric Validity: While the experiments are strong, exploring how BRIDGE performs with other RL algorithms or with larger datasets could strengthen generality claims. Additionally, the use of approximate co-influence metrics is intriguing but could benefit from validation or sensitivity analysis, especially if intended as a general-purpose diagnostic tool.

---

> ### Author Rebuttal · Authors · 2025-07-31
>
> We thank the reviewer for your thoughtful and constructive feedback. Here are our responses:
> - **W1 & Q1. How well does BRIDGE generalize to domains beyond math and structured reasoning tasks with programmatically verifiable rewards? How well does behavior injection generalize to more complex or interactive tool-use tasks beyond static QA or math problems?**
>
>   Thank you for the insightful questions. BRIDGE injects behaviors based on a DAG representation. As long as an underlying graph structure can be identified (e.g., from the answer CoT), which holds for many reasoning tasks, our method can generalize beyond math or structured QA domains. However, BRIDGE is mainly designed to enhance a model’s ability to leverage reasoning behaviors, such as decomposition and reflection, rather than injecting domain-specific knowledge. Therefore, while it can be applied to other domains, the improvements generally may be less pronounced in tasks where domain expertise is the primary challenge instead of multi-step reasoning (e.g., the non-reasoning tasks).
>
>   Regarding more complex or interactive tool-use tasks, we can still adapt our method by identifying a reasoning graph structure in the action sequence or interaction flow. While we believe exploring how BRIDGE operates in such interactive settings is an exciting direction, we consider it beyond the current scope and leave it to future work.
>
> - **W2. Manual Behavior Design: The effectiveness of BRIDGE depends on hand-crafted behaviors (e.g., reflection, subgoaling) tailored to specific tasks. This may limit generalization unless clearer guidelines or semi-automated methods for behavior selection are provided.**
>
>   Thank you for highlighting this important point. While the behaviors may appear hand-crafted, they are in fact designed based on general exploration and exploitation principles from RL, making them broadly applicable beyond the specific tasks studied in this paper. Our ablation results in fig.5 further demonstrate that all these behaviors contribute to the model’s performance during RL training.
>
>   Moreover, the specific behaviors we adopt such as reflection and subgoaling are grounded in well-established findings from cognitive science [1] and have also been shown to be effective in LLM finetuning [2]. We agree that developing more systematic or semi-automated methods for behavior selection would be a valuable direction for future work, and we see our current design as a first step toward that goal.
>
> - **W3. Base Model Dependence: BRIDGE assumes a reasonably capable base model that can already express chain-of-thought reasoning.**
>
>   We respectfully disagree with the assumption that BRIDGE requires a base model with strong reasoning capabilities. In our experiments, we intentionally use pretrained base models (Qwen2.5, Llama3) rather than the reasoning-focused post-training model (e.g., Qwen2.5-math or DeepSeek-distilled model). In fact, BRIDGE is designed to teach reasoning behaviors to models that initially lack such capabilities. By injecting structured behaviors like subgoaling and reflection, our method guides the model toward improved reasoning during both SFT and RL stages.
>
>   Our empirical results also validate: 1) The baseline models without behavior injection show limited gains from SFT and RL, indicating their initial lack of strong reasoning ability. 2) BRIDGE significantly boosts performance in the RL stage, demonstrating its effectiveness in enhancing reasoning in less capable base models.
>
> - **Q2. What’s the impact of the demonstration source or quality? Were they human-written, model-generated, or noisy? How robust is the method to imperfect traces?**
>
>   The demonstrations used in our experiments (i.e., the vanilla CoT) are rule-based and programmatically generated as part of the iGSM and PromptBench datasets. These ground-truth traces are consistent across all compared methods, including BRIDGE, PP-Aug, and RC-Aug. More implementation details can be found in Appendix B.1 and the codes.
>
>   Importantly, the focus of BRIDGE is not to evaluate or depend on the inherent quality of the base demonstrations, but rather to inject structured reasoning behaviors (e.g., subgoaling, reflection) into them. These behaviors are applied systematically and can be incorporated even in the presence of imperfect or noisy demonstrations. The core research question we address is whether behavior injection can improve RL fine-tuning effectiveness, regardless of the initial demonstration source. We agree that studying the robustness of BRIDGE to imperfect or low-quality traces is an important direction. While this is not the primary focus of the current paper, we consider it valuable future work and appreciate the suggestion.
>
> - **Q3. How does behavior injection compare to prompting approaches like “think step-by-step” at inference time? Does fine-tuning offer clear advantages?**
>
>   Yes, the finetuning shows clear advantages over prompt engineering. We evalute the accuracy on a iGSM test set (operation num=21~25) as follows
>
>   | Model     | Base (no SFT) | Base + prompting (no SFT) | Random Guess | BRIDGE (SFT) | BRIDGE (RFT) |
>   | --------- | ------------- | ---------------------- | ------------ | ------------ | ------------ |
>   | Qwen 1.5B | <10.0          | <10.0                   | 14.2         | 38.6         | 89.4         |
>   | Llama 1B  | <10.0          | <10.0                   | 14.2         | 27.4         | 57.4         |
>
>   The prompt engineering is not effective for synthetic questions as the base models are not familiar with this OOD tasks, which is even lower than random guess (random guess means the model answers "0" for all questions). As a comparison, the finetuning (SFT or RFT) shows much larger performance improvement.
>
> - **Q4. Are there cases where the model overuses tools unnecessarily? Any error analysis on when behavior injection might lead to misuse would be helpful.**
>
>   Based on our observations, the exploration behavior, particularly reflection, can indeed be overused by the SFT model trained with BRIDGE. In some cases, the model repeatedly attempts to solve an unlocked node without having computed all of its parent nodes, leading to output patterns such as:
>   ```
>   Then, let's denote the number of [Node] as y. But we haven't calculated the number of [one parent of the Node], thus the value of y is still unknown;
>   # repeat the above for several times
>   ```
>   This phenomenon is consistent with he reflection pattern injected during SFT, where the model learns to re-attempt unsolved steps. While such repetition reflects the intended exploration behavior, it can introduce errors and result in reduced performance at the SFT stage. However, as shown in fig. 5, although the model overuses reflection after SFT training, it will be corrected in further RL training with reward signals and those behaviors are finally beneficial to the performance after RL.
>
> - **Concerns on AI safety**
>
>   Thanks for your comprehensive discussion on the potential misuse of BRIDGE that can lead to the violation of AI safety. We totally understand and agree with your concerns. We will provide more discussions of the potential negative impact in revision.
>
> We sincerely appreciate your effort again and we hope our responses can address your concerns.
>
> [1] Polya, George. "How to solve it." (1957).
>
> [2] Gandhi, Kanishk, et al. "Cognitive behaviors that enable self-improving reasoners, or, four habits of highly effective stars."

---

> ### Comment · Area_Chair_vsNE · 2025-08-08
>
> Please explicitly respond to the author's rebuttal, in addition to the Mandatory Acknowledgement. Thanks!

---

### Official Review · Reviewer_JjKq · 2025-07-01

**Clarity:** 3
**Significance:** 3
**Originality:** 3
**Rating:** 5
**Confidence:** 4

**Summary:**

This paper presents BRIDGE (BehavioR Injection Data auGmEntation), a method to prepare language models for more effective reinforcement learning fine-tuning (RFT) by injecting exploration and exploitation behaviors during supervised fine-tuning (SFT). The core contributions of this paper are 1) analysis of reinforcement learning behavior and its correlation with rollout data; 2) proposed data augmentation strategy BRIDGE; 3) empirical evaluation. The authors identify two critical factors influencing RFT performance: rollout accuracy and data co-influence. Guided by this analysis, BRIDGE enhances SFT datasets by injecting exploration and exploitation behaviors, making models "RL-ready" and improving their learning efficiency and generalization in the RFT stage.

**Questions:**

- Why does BRIDGE yield quite low accuracy on SFT stage (Table 2)?
- Do authors scale up to 7B/32B models?
- How does the theory apply to other RL algorithms?
- What's the effect of training on BRIDGE-construcrted for more epoch? Will it affect the generalizability of the model?

**Ethical Concerns:**

["NO or VERY MINOR ethics concerns only"]

**Limitations:**

yes

**Quality:**

3

**Strengths And Weaknesses:**

Strength:
- Pre-RL preparation is important. Many prior literature [1,2] pin-point this but few works delve deep into the direction.
- The idea is well-motivated by theory
- Data efficiency. This paper shows a data-efficient preparation approach to incentivize language models' potentials in RL training, where the proposed strategy requires 2000/5000 data samples for two tasks according to the appendix.
- Remarkable empirical improvements.


Weakness:
- The general impact is upper-bounded by the scalability of DAG representation.
- The theoretical analysis is specific to GRPO, where the coefficient $\sqrt{\alpha(1-\alpha)}$ depends on GRPO's advantage estimation. It is unclear how the theory intuition will changes on other RL algorithms
- It is unclear how the training epochs will affect the BRIDGE performance. As shown in table 1-2, different epochs affect the final performance a lot but the effect of more training epochs is not presented.


[1] Guo, Daya, Dejian Yang, Haowei Zhang, Junxiao Song, Ruoyu Zhang, Runxin Xu, Qihao Zhu et al. "Deepseek-r1: Incentivizing reasoning capability in llms via reinforcement learning." arXiv preprint arXiv:2501.12948 (2025).
[2] Wang, Zengzhi, Fan Zhou, Xuefeng Li, and Pengfei Liu. "Octothinker: Mid-training incentivizes reinforcement learning scaling." arXiv preprint arXiv:2506.20512 (2025).

---

> ### Author Rebuttal · Authors · 2025-07-31
>
> We thank the reviewer for your thoughtful and constructive feedback. Here are our responses:
> - **W1. The general impact is upper-bounded by the scalability of DAG representation.**
>
>   Thank you for raising this important point. We agree that our method relies on the DAG representation. However, we believe that this representation is broadly applicable to many question types solvable by sequentially resolving intermediate steps within a CoT reasoning, which naturally contains a underlying DAG structure. To extract the DAG from more general questions, given a CoT-style answer, we can use an oracle LLM (e.g., GPT-4) to parse the structure by identifying intermediate variables, conclusions, or corollaries as nodes, and defining edges based on their dependencies. This allows us to construct a DAG even when the original question does not explicitly follow a fixed template.
>
>   Here is an example. We use the following prompt to extract DAG:
>   ```
>   You are a helpful data analyst. You will be given a question-answer pair. Your task is to extract the DAG of the question. Here are some instructions for extracting the DAG:
>   - First separate the answer into multiple steps. If one step includes multiple intermediate variables / conclusions, further separate it until each step includes only one.
>   - Identify NODE in each step. The node can be intermediate variables / conclusions / corollary.
>   - Identify the dependency between the nodes.
>   - Remember that the final graph should be acyclic.
>
>   You should return the DAG in the following format:
>   - the name of the nodes, e.g., node 1: x, node 2: y, node 3: z
>   - the dependent list of each node, e.g., node 1: [], node 2: [1], node 3: [1, 2]
>
>   [few-shot examples]
>   [QA]
>   ```
>   Given a QA sample with the highest difficulty (level 5) from MATH dataset:
>   ```
>   Question:
>   Define
>   \[
>   p = \sum_{k = 1}^\infty \frac{1}{k^2} \quad \text{and} \quad q = \sum_{k = 1}^\infty \frac{1}{k^3}.
>   \]
>   Find a way to write
>   \[
>   \sum_{j = 1}^\infty \sum_{k = 1}^\infty \frac{1}{(j + k)^3}
>   \]
>   in terms of $p$ and $q$.
>
>   Answer:
>   We count the number of times $\frac{1}{n^3}$ appears in the sum
>   \[
>   \sum_{j = 1}^\infty \sum_{k = 1}^\infty \frac{1}{(j + k)^3},
>   \]
>   where $n$ is a fixed positive integer. (In other words, we are conditioning the sum on $j + k$.)
>   We get a term of $\frac{1}{n^3}$ each time $j + k = n.$ The pairs $(j,k)$ that work are $(1,n - 1),$ $(2,n - 2),$ $\dots,$ $(n - 1,1),$ for a total of $n - 1$ pairs.
>   Therefore,
>   \begin{align*}
>   \sum_{j = 1}^\infty \sum_{k = 1}^\infty \frac{1}{(j + k)^3} &= \sum_{n = 1}^\infty \frac{n - 1}{n^3} \\
>   &= \sum_{n = 1}^\infty \left( \frac{n}{n^3} - \frac{1}{n^3} \right) \\
>   &= \sum_{n = 1}^\infty \left( \frac{1}{n^2} - \frac{1}{n^3} \right) \\
>   &= \sum_{n = 1}^\infty \frac{1}{n^2} - \sum_{n = 1}^\infty \frac{1}{n^3} \\
>   &= \boxed{p - q}.
>   \end{align*}
>   ```
>   The extract result is:
>   ```
>   Node names:
>   node 1: ( S = \sum_{j = 1}^\infty \sum_{k = 1}^\infty \frac{1}{(j + k)^3} )
>   node 2: Number of pairs ((j, k)) with (j + k = n) is ( n-1 )
>   node 3: ( S = \sum_{n=1}^\infty \frac{n-1}{n^3} )
>   node 4: ( \frac{n-1}{n^3} = \frac{1}{n^2} - \frac{1}{n^3} )
>   node 5: ( S = \sum_{n=1}^\infty \left(\frac{1}{n^2} - \frac{1}{n^3}\right) )
>   node 6: ( S = \sum_{n=1}^\infty \frac{1}{n^2} - \sum_{n=1}^\infty \frac{1}{n^3} )
>   node 7: ( S = p - q )
>
>   Dependency list:
>   node 1: []
>   node 2: []
>   node 3: [1, 2]
>   node 4: []
>   node 5: [3, 4]
>   node 6: [5]
>   node 7: [6]
>   ```
>   This indicates that we can extract DAG representation from other QA data and our method is able to generalize to a broad range of tasks.
>
> - **W2 & Q3. The theoretical analysis is specific to GRPO, where the coefficient depends on GRPO's advantage estimation... How does the theory apply to other RL algorithms?**
>
>   Thanks for this question. The key of applying the theory to other RL algorithm is to replace the advantage function. Following the notation in our original paper, i.e., given a query with $n$ correct outputs and $N-n$ wrong outputs, the accuracy $\alpha = n/N$ and the advantages of correct and wrong output are $A_+, A_-$ respectively. We consider three RL variants:
>   - Dr.GRPO [1]. The advantages are $A_+ = 1-\frac{n}{N}, A_- = -\frac{n}{N}$. Plug-in them to Eq.(10) (in appendix), the corresponding gradient is $$
>     \nabla_\theta \mathcal{J} = \mathbb{E} \frac{n(N-n)}{N^2}[\frac{1}{n}\sum_{i=1}^n \log\pi_\theta(o_{i+}|q) - \frac{1}{N-n}\sum_{j=1}^{N-n}\log\pi_\theta(o_{j-}|q)] \\
>     =  \mathbb{E} \alpha(1-\alpha) [...]
>     $$ Then its per-step influence is to replace the original coefficient $\sqrt{\alpha(1-\alpha)}$ by $\alpha(1-\alpha)$ and other parts remain the same as GRPO.
>   - GPG [2]. It multiples the advantage by a coefficient $C$ (it is $\alpha$ in original paper and we use $C$ instead to avoid ambiguity). The advantages are $A_+ = C\cdot(1-\frac{n}{N}), A_- = C\cdot (-\frac{n}{N})$. Similar to Dr.GRPO, the coefficient in per-step influence is $C \alpha (1-\alpha)$ and other parts remain the same.
>   - DAPO [3]. The advantage computation in DAPO is the same as the GRPO so the coefficient is still $\sqrt{\alpha (1-\alpha)}$. DAPO introduces other modifications such as query filtering. Therefore, the corresponding per-step influence should replace the original query set $Q$ by a filtered query set $Q'$ which only includes queries with rollout accuracy $\in(0,1)$.
>
>   The results on other RL variants show that "section 3.2 takeaway" holds for different RL algorithms, i.e., the rollout accuracy and co-influence are still two key factors to the performance gain in RL.
>
> - **W3 & Q4. ... As shown in table 1-2, different epochs affect the final performance a lot but the effect of more training epochs is not presented. What's the effect of training on BRIDGE-construcrted for more epoch? Will it affect the generalizability of the model?**
>
>   We want to first clarify that the performances in table 1 and 2 are not comparable: table 1 shows the results on iGSM task while table 2 shows the results on PromptBench. They have different SFT dataset sizes. Moreover, the SFT epoch only affects the SFT training and we adopt the exact same settings for the follow-up RL training.
>
>   We provide the results of training with different SFT epochs on iGSM tasks (where we adopt epoch num = 5 in original paper). The following table shows the accuracy on validation set (the same as the set in fig.5).
>
>   | Base      | Epoch num | 2    | 5    | 10   | 20   |
>   | --------- | --------- | ---- | ---- | ---- | ---- |
>   | Qwen 1.5B | SFT       | 38.4 | 38.6 | 39.4 | 36.0 |
>   |           | RFT       | 86.8 | 89.4 | 88.2 | 77.6 |
>   | Llama 1B  | SFT       | 20.4 | 27.4 | 28.6 | 24.0 |
>   |           | RFT       | 55.2 | 57.4 | 54.6 | 40.4 |
>
>   In general, more SFT epoch slightly improves the performance of the SFT model when epoch num $\leq 10$. The performance of RL model is not very sensitive to the SFT epoch in the range of 2 ~ 10. However, we also observe a performance drop if training SFT for 20 epochs. The main reason is that the excessive SFT training decreases the entropy of LLM output, which reduces the exploration in RL training and leads to slower performance increase.
>
> - **Q1. Why does BRIDGE yield quite low accuracy on SFT stage (Table 2)?**
>
>   That's a good question. The relatively low accuracy of the SFT model mainly stems from over-reflection after BRIDGE injects reflection behaviors. Specifically, the model imitates the injected reflection pattern in the BRIDGE SFT data and repeatedly attempts to solving an unlocked node (i.e., a node whose parent nodes have not yet been resolved) and then reflects. This often leads to unnecessarily long reasoning chains that reach the generation length limit, causing the model to fail to complete the answer and identified as wrong.
>
>   While this behavior results in low initial accuracy, it is effectively corrected during RL fine-tuning. As shown in our results, RL training significantly improves its final performance.
>
> - **Q2. Scale up to 7B/32B models**
>
>   According to reviewer's suggestion, we ran the experiments on iGSM tasks with Qwen2.5-7B and llama3.1-8B models. The settings are the same as the previous experiments: we first train the base model by SFT for 5 epochs and then run RL for 100 steps.
>
>   | Base     |    | Vanilla     | PP-Aug      | RC-Aug      | BRIDGE              |
>   | -------- | -- | --- | ---- | --- | ---- |
>   | Qwen 7B  | SFT      | 54.4 / 37.6 | 61.8 / 41.2 | 53.4 / 33.8 | 61.8 / 49.4         |
>   |          | RFT      | 74.2 / 64.8 | 76.0 / 62.0 | 73.0 / 60.8 | **91.0** / **83.2** |
>   |          | $\Delta$ | 19.8 / 27.2 | 14.2 / 20.8 | 19.6 / 27.0 | **29.2** / **33.8** |
>   | LLama 8B | SFT      | 61.6 / 50.8 | 69.0 / 57.8 | 60.8 / 43.8 | 53.0 / 46.4         |
>   |          | RFT      | 68.4 / 59.2 | 76.4 / 62.8 | 65.6 / 57.8 | **89.4** / **87.6** |
>   |          | $\Delta$ | 6.8 / 8.4   | 7.4 / 5.0   | 4.8 / 14.0  | **36.4** / **41.2** |
>
>   We can observe that BRIDGE achieves the best performance with 7B and 8B models, which shows the scalability of our method. For tasks that require capability of 32B+ model size, we leave them as future works due to our current constraints on computation resources.
>
> We sincerely appreciate your effort again and we hope our responses can address your concerns.
>
>
> [1] Understanding R1-Zero-Like Training: A Critical Perspective
>
> [2] GPG: A Simple and Strong Reinforcement Learning Baseline for Model Reasoning
>
> [3] DAPO: An Open-Source LLM Reinforcement Learning System at Scale

---

### Official Review · Reviewer_NxGR · 2025-07-09

**Clarity:** 3
**Significance:** 3
**Originality:** 3
**Rating:** 5
**Confidence:** 4

**Summary:**

This paper proposes BehavioR Injection Data auGmEntation (BRIDGE), a strategy designed to augment the training data for SFT in order to improve the performance of LLMs trained via RL. Since RL training is typically preceded by SFT, BRIDGE enhances the SFT dataset by constructing examples that incorporate both exploration and exploitation behaviors, thereby better preparing LLMs for subsequent RL training. Experiments are conducted on two families of LLMs: Qwen and Llama, and evaluated on two benchmarks: iGSM and PromptBench. The results show that LLMs trained on datasets augmented with BRIDGE achieve significant performance improvements after RL training, outperforming models trained with other data augmentation strategies.

**Questions:**

1. Could you provide more details on how to extract the DAG representation from a query and its corresponding response?

2. As shown in Table 2, it seems that the experimental results for Qwen2.5-3B are missing from PromptBench.

3. Could you explain why the injected behaviors influence the accuracy distribution of the rollout samples?

**Ethical Concerns:**

["NO or VERY MINOR ethics concerns only"]

**Final Justification:**

5: Accept

**Limitations:**

yes

**Quality:**

3

**Strengths And Weaknesses:**

> **Strengths**

1. BRIDGE is motivated by per-step influence analysis, which highlights two key factors for enhancing the performance of LLMs: (1) rollout accuracy and (2) the co-influence coefficient between training and target data.

2. Experimental results show that LLMs trained on data augmented with BRIDGE achieve exceptional performance after RL training.

3. In addition to experiments demonstrating the effectiveness of BRIDGE-augmented data, the paper includes ablation studies that analyze various injected behaviors. These studies provide valuable insights into constructing SFT datasets to improve LLM performance following RL training.

> **Weaknesses**

1. The LLMs used in the experiments have no more than 3 billion parameters. It would be preferable to include experimental results on more widely used LLMs in academic research, such as those with approximately 7 billion parameters.
2. Considering that the Adam optimizer is predominantly used for training LLMs, the mathematical formulation for estimating the per-step influence of q on model performance may lack accuracy.

---

> ### Author Rebuttal · Authors · 2025-07-31
>
> We thank the reviewer for your thoughtful and constructive feedback. Here are our responses:
> - **W1. Include experimental results on larger models (such as 7B).**
>
>   We run experiments on iGSM tasks with Qwen2.5-7B and llama3.1-8B models. The settings are the same as the previous experiments: we first apply SFT for 5 epochs on the base model and then run RL for 100 steps. Here are the results of their accuracy (the numbers before and after the `/` indicate the accuracy on in-domain and OOD test set respectively).
>
>   | Base     |    | Vanilla     | PP-Aug      | RC-Aug      | BRIDGE   |
>   | --- | --- | --- | --- | --- | ---- |
>   | Qwen 7B  | SFT      | 54.4 / 37.6 | 61.8 / 41.2 | 53.4 / 33.8 | 61.8 / 49.4         |
>   |          | RFT      | 74.2 / 64.8 | 76.0 / 62.0 | 73.0 / 60.8 | **91.0** / **83.2** |
>   |          | $\Delta$ | 19.8 / 27.2 | 14.2 / 20.8 | 19.6 / 27.0 | **29.2** / **33.8** |
>   | LLama 8B | SFT      | 61.6 / 50.8 | 69.0 / 57.8 | 60.8 / 43.8 | 53.0 / 46.4         |
>   |          | RFT      | 68.4 / 59.2 | 76.4 / 62.8 | 65.6 / 57.8 | **89.4** / **87.6** |
>   |          | $\Delta$ | 6.8 / 8.4   | 7.4 / 5.0   | 4.8 / 14.0  | **36.4** / **41.2** |
>
>   These results demonstrate that BRIDGE maintains its significant advantage over all baselines on these larger models. The substantial improvement from RL (Δ) when using BRIDGE suggests that our method effectively equips the base model with generalizable reasoning capabilities, making it more amenable to reinforcement learning, rather than simply encouraging it to memorize specific training examples. We believe these new results substantially strengthen our paper's claims.
>
> - **W2. The estimating the per-step influence of q on model performance may lack accuracy due to the Adam optimizer.**
>
>   Thank you for raising this important concern. We acknowledge that estimating per-step influence under adaptive optimizers like Adam introduces certain approximations. However, we would like to offer a few points of clarification: 1) The estimation is particularly accurate at the initial steps of training, where the Adam optimizer behaves similarly to vanilla SGD. Since the performance gain in RL is mainly in the early stage of the training, we find the approximation still meaningful and informative. 2) More faithful modeling of Adam optimizer requires tracking its state, which is computationally intractable in our large-scale RL setting. Previous work [1] attempts to estimate this state using separate data, but such approaches also introduce additional approximation. 3) Our approach aligns with prior studies (e.g., [2]), which also adopts SGD approximations when analyzing training dynamics under Adam. Such approximations have been shown to offer valuable insights without optimizer state.
>
>   Overall, while we agree that more precise modeling of per-step influence under Adam is an interesting direction, we believe our current approximation still provides a reasonable and useful estimate. We appreciate the suggestion and consider this a valuable avenue for future work.
>
> - **Q1. More details on how to extract the DAG representation from a query and its corresponding response**
>
>   Thanks for your question. we are happy to provide more details here.
>   - For the datasets we used in this paper (iGSM and promptbench),  all questions and answers are rule-based and follow a fixed sentence structure, which allows us to extract the DAG representation through simple **string matching**. Take a iGSM problem for example, each variable mentioned in the question is treated as a node, and we define an edge from one node to another if the value of the former depends on the latter. Consider a premise `The number of each Painting Room’s Printed Casual Backpack equals 9 more than the sum of each Painting Room’s Designer Bag and each Pottery Classroom’s Printed Casual Backpack.`. we view `each Painting Room’s Printed Casual Backpack` as a node. Since it depends on nodes `each Painting Room’s Designer Bag` and `each Pottery Classroom’s Printed Casual Backpack` and we draw edges from these two nodes to the former. By iterating all premises in this way, we construct the node set V and edge set E of the graph. Similarly, we can also apply the same procedure to the answer. However, we will miss the redundant nodes (which are defined in the question but do not contribute to the computation of the final node, e.g., the gray node in fig.7 in appendix) if the answer only includes the minimal topological path to the final node.
>   - For other QA datasets without a fixed format, we can employ an oracle LLM (e.g., GPT4) to parse the node and edge from the QA. Specifically, we can view all intermediate variables / conclusions / corollaries as node and the edge is still their dependency. Here is a prompt to extract (V, E):
>     ```
>     You are a helpful data analyst. You will be given a question-answer pair. Your task is to extract the DAG of the question. Here are some instructions for extracting the DAG:
>     - First separate the answer into multiple steps. If one step includes multiple intermediate variables / conclusions, further separate it until each step includes only one.
>     - Identify NODE in each step. The node can be intermediate variables / conclusions / corollary.
>     - Identify the dependency between the nodes.
>     - Remember that the final graph should be acyclic.
>
>     You should return the DAG in the following format:
>     - the name of the nodes, e.g., node 1: x, node 2: y, node 3: z
>     - the dependent list of each node, e.g., node 1: [], node 2: [1], node 3: [1, 2]
>
>     [few-shot examples]
>     [QA]
>     ```
>     Given a QA sample from MATH dataset:
>     ```
>     Question:
>     Define
>     \[
>     p = \sum_{k = 1}^\infty \frac{1}{k^2} \quad \text{and} \quad q = \sum_{k = 1}^\infty \frac{1}{k^3}.
>     \]
>     Find a way to write
>     \[
>     \sum_{j = 1}^\infty \sum_{k = 1}^\infty \frac{1}{(j + k)^3}
>     \]
>     in terms of $p$ and $q$.
>
>     Answer:
>     We count the number of times $\frac{1}{n^3}$ appears in the sum
>     \[
>     \sum_{j = 1}^\infty \sum_{k = 1}^\infty \frac{1}{(j + k)^3},
>     \]
>     where $n$ is a fixed positive integer. (In other words, we are conditioning the sum on $j + k$.)
>     We get a term of $\frac{1}{n^3}$ each time $j + k = n.$ The pairs $(j,k)$ that work are $(1,n - 1),$ $(2,n - 2),$ $\dots,$ $(n - 1,1),$ for a total of $n - 1$ pairs.
>     Therefore,
>     \begin{align*}
>     \sum_{j = 1}^\infty \sum_{k = 1}^\infty \frac{1}{(j + k)^3} &= \sum_{n = 1}^\infty \frac{n - 1}{n^3} \\
>     &= \sum_{n = 1}^\infty \left( \frac{n}{n^3} - \frac{1}{n^3} \right) \\
>     &= \sum_{n = 1}^\infty \left( \frac{1}{n^2} - \frac{1}{n^3} \right) \\
>     &= \sum_{n = 1}^\infty \frac{1}{n^2} - \sum_{n = 1}^\infty \frac{1}{n^3} \\
>     &= \boxed{p - q}.
>     \end{align*}
>     ```
>     The extract result is:
>     ```
>     Node names:
>     node 1: ( S = \sum_{j = 1}^\infty \sum_{k = 1}^\infty \frac{1}{(j + k)^3} )
>     node 2: Number of pairs ((j, k)) with (j + k = n) is ( n-1 )
>     node 3: ( S = \sum_{n=1}^\infty \frac{n-1}{n^3} )
>     node 4: ( \frac{n-1}{n^3} = \frac{1}{n^2} - \frac{1}{n^3} )
>     node 5: ( S = \sum_{n=1}^\infty \left(\frac{1}{n^2} - \frac{1}{n^3}\right) )
>     node 6: ( S = \sum_{n=1}^\infty \frac{1}{n^2} - \sum_{n=1}^\infty \frac{1}{n^3} )
>     node 7: ( S = p - q )
>
>     Dependency list:
>     node 1: []
>     node 2: []
>     node 3: [1, 2]
>     node 4: []
>     node 5: [3, 4]
>     node 6: [5]
>     node 7: [6]
>     ```
>     This example shows that we can still extract DAG from other QA data without a pre-defined format.
>
>   We will update the details and examples in the revised manuscript.
>
> - **Q2. In Table 2, the results for Qwen2.5-3B are missing from PromptBench.**
>
>   Thanks for your reminder. We ran the experiment on Qwen2.5-3B on PromptBench. Here are the results (RL is trained for 100 steps):
>
>   | Base    |   | Vanilla     | PP-Aug      | RC-Aug      | BRIDGE  |
>   | --- | --- | --- | ---- | ---- | ---- |
>   | Qwen 3B | SFT      | 31.2 / 18.0 | 40.6 / 29.4 | 35.2 / 19.4 | 44.4 / 31.0         |
>   |         | RFT      | 50.0 / 34.6 | 66.0 / 50.8 | 64.0 / 43.4 | **85.8** / **70.0** |
>   |         | $\Delta$ | 18.8 / 16.6 | 25.4 / 21.4 | 28.8 / 24.0 | **41.4** / **39.0** |
>
>   In the results, we can observe that BRIDGE achieves the best performance among all the baselines, which is consistent to the results we presented in the main context.
>
> - **Q3. Why the injected behaviors influence the accuracy distribution of the rollout samples?**
>
>   Because the training data (with and without behavior injection) differs in both content and structure, which naturally leads to differences in the resulting SFT models’ performance. Specifically, there are two main contributing factors: 1) Exploitation behaviors often improve accuracy, as they encourage the model to better utilize the available information, reinforcing effective problem-solving strategies. 2) Exploration behaviors, on the other hand, may reduce accuracy of the SFT model. This is because they deliberately introduce incorrect intermediate reasoning steps (e.g., through reflection), which can cause the model to make errors on questions it would otherwise solve correctly.
>
>   While these behaviors have different effects on the SFT model’s accuracy, they are all valuable from an RL training perspective. As shown in Fig. 5, incorporating both exploration and exploitation behaviors improves the overall performance of the RL-finetuned model, suggesting that behavior injection plays a crucial role in enhancing the model’s learning dynamics.
>
> We sincerely appreciate your effort again and we hope our responses can address your concerns.
>
> [1] Xia, Mengzhou, et al. "Less: Selecting influential data for targeted instruction tuning."
>
> [2] Ren, Yi, and Danica J. Sutherland. "Learning dynamics of llm finetuning."

---

> > ### Comment · Reviewer_NxGR · 2025-08-09
> >
> > I thank the authors for their detailed responses and the additional experiments. The new results further demonstrate the effectiveness of BRIDGE. I believe BRIDGE represents a valuable contribution to the research community, and I would be pleased to see this paper accepted. Since a rating of 5 reflects a positive assessment, I will maintain my score.

---

### Author Response · Authors · 2025-08-07
**Reminder of the discussion**

Hi reviewers,

We would like to express our gratitude for your valuable feedback. As the discussion phase approaching the end, we want to draw your attention to our rebuttal and new results to address your concerns. We are also willing to discuss more if you have other questions. We would greatly appreciate a response and your reconsideration our manuscript in light of our clarifications.

---

### Decision · Program_Chairs · 2025-09-17

**Decision:**

Accept (poster)

**Comment:**

This paper tackles the open question of why reinforcement fine-tuning (RFT) benefits some language models but not others, and proposes a simple yet powerful solution: behavior injection (BRIDGE), a data-centric augmentation that seeds exploration and exploitation behaviors during supervised fine-tuning to make models more “RL-ready.” The reviewers appreciated the combination of careful analysis (identifying rollout accuracy and data co-influence as key factors in RFT effectiveness) with a practical method that is broadly applicable. Empirical results across math and logical reasoning benchmarks show consistent and significant improvements over strong augmentation baselines, with ablations confirming the role of injected behaviors. While some reviewers noted that the experiments are focused on relatively narrow domains and the theoretical analysis is limited in scope, the authors clarified these concerns in discussion and positioned their method as a general data-centric approach. Overall, the paper offers both useful conceptual insights and strong empirical gains, and I recommend acceptance.